# Positive attitudes towards feline obesity are strongly associated with ownership of obese cats

Kendy T. Teng[1,2]*, Paul D. McGreevy[1], Jenny-Ann L. M. L. Toribio[1], Navneet K. Dhand[1]

**1** Sydney School of Veterinary Science, Faculty of Science, University of Sydney, Camperdown, NSW, Australia, **2** VISAVET Health Surveillance Centre, Complutense University of Madrid, Madrid, Spain

* kendy.t.teng@gmail.com

**Data Availability Statement:** The data are now in DOI:10.5281/zenodo.3852412

**Funding:** KTT has received 2015 RSPCA Australia Alan White Scholarship for Animal Welfare from

## Abstract

Overweight and obesity (O&O) is a risk factor for several health conditions and can result in a shorter lifespan for cats. The objectives of this study were to investigate (a) cat owners' attitudes towards feline O&O and their associations with O&O in their cats; and (b) the risk factors for feline O&O and underweight, particularly those involving owner practice. An online survey comprising questions related to cat owners' attitudes towards feline O&O, owner-reported body weight and body condition of their cat, and potential risk factors for feline O&O was conducted. Primarily targeting the Australian population, the survey attracted 1,390 valid responses. In response to ten attitude-related questions, more participants (percentage range among the ten questions: 39.1–76.6%) held a disapproving attitude towards feline O&O than a neutral (17.1–31.9%) or approving attitude (3.9–27.7%). A greater proportion of participants had a more disapproving attitude towards obesity than towards overweight. Cats belonging to owners with an approving attitude towards O&O were more likely to be overweight or obese than cats belonging to owners with a disapproving attitude towards O&O. The cats had particularly high odds of overweight or obesity if their owner agreed that 'being chubby says that the cat has a quality life' (OR: 3.75, 95% CI: 2.41–5.82) and 'being fat says that the cat has a quality life' (OR: 4.98, 95%CI: 2.79–8.91). This study revealed, for the first time, that begging for food was a risk factor for O&O in cats. Other important feline risk factors for O&O identified included being middle-aged, being mixed-breed, dry food as the major diet, the amount of feed not being quantified, and frequently spending time indoors. Being over 11 years, receiving no dry food and receiving measured amounts of feed were associated with an increased odds of underweight in cats. As specific attitudes often lead to certain behaviours, reducing approving attitudes towards feline O&O may potentially reduce the frequency of O&O and the risks of O&O-related disorders in cats.

Royal Society for the Prevention of Cruelty to Animals (RSPCA) Australia to conduct the current study. The RSPCA Australia plays no role in the study design, data collection and analysis, decision to publish, or preparation of the manuscript.

**Competing interests:** The authors have declared that no competing interests exist.

## Introduction

Research on overweight and obesity (O&O) has extended from humans to companion animals, with increasing recognition of the issue of O&O as a risk to health conditions, a shorter lifespan and impaired welfare of cats and dogs [1–5]. Studies investigating the risk factors for feline O&O can be categorised into two groups. Some focus on the cats themselves (i.e., intrinsic or host-related risk factors, such as breed, sex and age), whereas others explore aspects external to the cat (i.e., extrinsic or environment-related risk factors). Most studies investigating extrinsic risk factors for feline O&O have focused on owners' management of their cats. In contrast to intrinsic risk factors that have been well documented in the past 30 years, extrinsic risk factors have shown relatively inconsistent associations. Details of risk factors investigated are shown in S1 Table. Briefly, many studies have shown that male sex [6–10], neutered cats [11–17], middle age [6, 11–13, 18–21] and mixed breed [12, 14, 19–21] are associated with an increased risk of O&O. The extrinsic risk factors supported by the best evidence include feeding dry food [9, 22] and feeding treats/table scraps [6, 9, 15].

A largely unexplored owner attribute that may influence feline O&O is the owners' perception of their relationship with their cat. That said, one study that has specifically investigated this found that feline O&O was more likely among cats whose owners showed more affection to cats and over-humanised them [23]. In contrast, owner perception of cat body condition has been examined by several studies, with owner underestimation of cat body condition being linked consistently to a higher body condition score (BCS) in cats [8–10]. Another owner attribute potentially related to feline O&O is their attitude towards feline O&O. In humans, obese and overweight individuals are often stigmatised and may even suffer discrimination [23]. However, anecdotally, this outcome does not appear to be replicated in cats. Clearly, overweight and obese cats would not be judged to be lacking in self-discipline as occurs for overweight and obese humans. Instead, some people seem to have a positive attitude towards feline O&O and even relate chubbiness and fatness with cuteness in cats [24, 25]. As certain attitudes are drivers for certain behaviours [26], the attitude towards feline O&O among cat owners likely affects how they feed and interact with their cats, which can affect the body condition of the cats. An example in humans is the parental anti-fat attitude that has been shown to predict the application of restrictive feeding on their children to prevent them from becoming overweight or obese [27]. It is possible that cat owners' attitudes towards (feline) O&O can be a risk factor for O&O in cats as cats are increasingly being regarded as children or family members [28, 29].

To provide greater evidence for potential extrinsic risk factors for cat O&O, and to improve our understanding of the attitudes towards feline O&O among cat owners and its associations with their cats' body condition, the current study was conducted. Specifically, it investigated (a) the risk factors for owner-assessed feline O&O and underweight, particularly those involving owner practice and (b) owners' attitudes towards feline O&O and their associations with O&O in their cats. We hypothesised that cats with owners who have a more approving or even complimentary attitude towards feline O&O are more likely to be overweight or obese.

## Methods

### Ethics statement and funding sources

The ethics approval for this study was given by the University of Sydney Human Research Ethics Committee (Approval number: 2016/804). The project was funded by the Royal Society for the Prevention of Cruelty to Animals Australia (RSPCA Australia).

## Study design

This survey was conducted as part of the Australian Pet Welfare Survey. Assuming that 50% of the participants had a positive attitude towards feline O&O, a sample size of 1,068 cat owners was required to estimate the attitude of the participants with 95% confidence and 3% precision. Participants in the study were required to be over 18 years old, be fluent in English and own at least one cat. There was no restriction on the geographic location/extent of the target population for this survey.

**Questionnaire design and implementation.** The questionnaire (S1 File) had five sections. Section 1 focused on detailed cat demographics. Section 2 featured questions related to the cat's body weight and body condition. Three BCS measures were asked in this section: 1) participant-perceived BCS designated by participants based on descriptions in the questionnaire ('BCS Owner'), 2) BCS that was determined by participants choosing from five different depictions of cat shape the one most similar to their cat's shape ('BCS Figure'), and 3) BCS determined by the cat's veterinarians in the past year ('BCS Vet'). The BCS of 1-to-5 for BCS Owner and BCS Vet were labelled as: 'very underweight (BCS of 1)', 'somewhat underweight (BCS of 2)', 'ideal (BCS of 3)', 'chubby/overweight (BCS of 4)', 'fat/obese (BCS of 5)'. Section 3 contained questions associated with ownership, particularly with how the cat was fed, the lifestyle of the cat, and the interactions between the participant and the cat. Section 4 explored the participants' attitudes towards feline O&O. Ten questions asked about participant attitudes, five about attitudes to feline overweight and five about attitudes to feline obesity (Table 3; question 30 in S1 File). In the questionnaire, the terms 'chubby' and 'fat' were used as proxies for overweight and obesity, respectively. Each question provided five options (strongly disagree, disagree, neutral, agree and strongly agree). Lastly, Section 5 sought general demographic information about the participant. Participants with more than one cat were asked to answer the questions regarding the first cat when their names were listed alphabetically.

The questionnaire used an online interface provided by SurveyMonkey (SurveyMonkey Inc., San Mateo, California, USA). The survey was open for two months from 1st Nov 2016 to 31st Dec 2016 and was promoted through various ways such as online posting, sending leaflets to veterinary clinics, animal charities and advertising the survey on social media. The advertisements did not mention overweight or obesity but did mention the aim of the study (S2 File). Ten AUD50 gift cards were advertised as the incentive to increase the response rate.

## Statistical analyses

**Data management.** Data cleaning and management were undertaken in Microsoft Excel (Microsoft Corp. Redmond, Washington, United States) and R version 3.3.0 (R Core Team) with RStudio interface (RStudio Team), facilitated by the 'car' [30] and 'plyr' [31] packages. All the analyses were conducted in RStudio.

A response was included in the analysis only if the participant answered at least one of the questions about the evaluation of the BCS of their cat. For Australian residential participants who provided their postcode, participants were classified as living in 'urban' or 'rural' areas by consulting information from a marketing website [32].

Two main analyses were conducted: the first examined the associations between the owners' attitude towards feline O&O and the owner-reported BCS of their cats; the second investigated the risk factors for feline O&O and underweight by using multinomial logistic regression. The significance level was set at $P<0.05$ throughout this study unless indicated otherwise.

**BCS (outcome variable).** The three candidates for the cat BCS outcome variable were BCS Owner, BCS Figure, and BCS Vet. Although BCS Vet was most likely to be close to the

true BCS of cats, it could not be considered because many participants (n = 571, 41.1%) did not provide this information. To determine whether BCS Owner or BCS Figure was more suitable, the levels of agreement between BCS Vet and both BCS Owner and BCS Figure were evaluated separately by calculating weighted kappa using the 'psych' package [33]. Furthermore, the weighted kappa between categorised body weight and BCS Owner and BCS Figure were calculated for several breeds of cats whose ideal weight ranges were documented on a website named 'Cat Owner Club' [34]. BCS Owner [with three categories: underweight (BCS of 1 or 2), ideal weight (BCS of 3) and O&O (BCS of 4 or 5)] was chosen as the outcome variable for the analyses reported here because the values of weighted kappa between BCS Owner and both BCS Vet and categorised body weight from Cat Owner Club were higher than those between BCS Figure and these two variables.

**The association of the participants' attitude towards owner-estimated feline O&O with their cat's body condition.** Answers for each of the ten questions were used to classify each of the participants as having an (a) approving attitude (towards overweight or obesity), (b) neutral attitude or (c) disapproving attitude. Using owner-assessed cats' body condition (i.e., ideal and O&O) as the binomial outcome and the responses to ten questions about participants' attitudes as explanatory variables, binomial logistic regression was conducted in R to examine their associations. Underweight cats were excluded because the attitude questions were not about underweight in cats. Multivariable logistic regression was not conducted because the explanatory variables were clustered in two groups and were not independent. Two questions: "Being chubby is a disease" and "Being fat is a disease" were excluded in the analysis due to the undefined state of disease for O&O in (veterinary) medicine [35].

**Analyses of the risk factors for feline overweight and obesity and underweight.** *Explanatory variable management.* The explanatory variables included were grouped into 'cat demographic-related factors (Table 1)', 'feeding-related factors (S2 Table)', 'activity-related factors (S3 Table)' and 'participant-demographic-related factors (S4 Table)'. Some potential confounders of participants' demographics (i.e., *age*, *gender*, *education level*, *being a veterinarian* and *animal-related profession*) were included in the analyses (S3 Table). All the variables in Table 1 and S2 Table–S4 Table were included in the analyses except for neuter status, due to all the intact cats being ideal-weight. Explanatory variables were re-categorised if, in the contingency table against outcome variable, more than 20% of the cells had less than five cats and if any of the cells of contingency tables contained a zero. Re-categorisation was performed to ensure no compromise of the biologically meaningful inference, as described in footnotes for Table 1 and S2 Table–S4 Table.

*Multinomial logistic regression analysis.* Multinomial logistic regression was conducted with the 'nnet' [36] and 'car' packages [30] in R. Univariable analyses identified unconditional relationships of the outcome variable with explanatory variables, which were included for multivariable model selection if they had a P-value less than 0.20. Variables with more than 15% of data missing were not considered for inclusion in the multivariable analysis. Collinearity between the explanatory variables was tested by Spearman's rank correlation coefficient with the 'Hmisc' package [37]. If a correlation coefficient between a pair of variables was greater than 0.7, only the variable with a stronger association with the outcome variable (i.e., smaller P-value in the univariable model) was retained for further analysis. A forward variable selection process was applied using the significant level as the criterion of inclusion. Pairwise interactions that might be biologically meaningful were evaluated in the model and, if significant, retained. Confounders were included in the final model if at least half of the coefficients of the variables in the final model, changed by more than 20% after the addition of the confounder in the model. Two separate models with binomial outcomes, underweight versus ideal weight and overweight versus ideal weight, with all the final explanatory variables included were fitted

**Table 1. Contingency tables of potential cat demographic risk factors for feline underweight or overweight and obesity with different body condition scores (BCS; 1 to 5) evaluated by 1,390 cat owners, based on data collected by an Australian-based online survey in 2016.**

| Variable | Category | BCS1 | BCS2 | BCS3 | BCS4 | BCS5 | Total | Grand total |
|---|---|---|---|---|---|---|---|---|
| Age | <1 year | 0 (0.0%) | 10 (5.2%) | 163 (84.0%) | 21 (10.8%) | 0 (0.0%) | 194 (14.0%) | 1,389 (99.9%) |
| | ≥1 to <3 years | 0 (0.0%) | 8 (2.6%) | 231 (76.2%) | 60 (19.8%) | 4 (1.3%) | 303 (21.8%) | |
| | ≥3 to <11 years | 1 (0.1%) | 29 (4.3%) | 436 (65.0%) | 187 (27.9%) | 18 (2.7%) | 671 (48.3%) | |
| | ≥11 years | 10 (4.5%) | 42 (19.0%) | 123 (55.7%) | 38 (17.2%) | 8 (3.6%) | 221 (15.9%) | |
| Breed | Mixed | 9 (0.8%) | 61 (5.5%) | 746 (67.1%) | 272 (24.5%) | 24 (2.2%) | 1,112 (80.0%) | 1,390 (100.0%) |
| | Pedigree | 2 (1.7%) | 17 (9.8%) | 128 (74.0%) | 22 (12.7%) | 4 (2.3%) | 173 (12.4%) | |
| | Purebred | 0 (0.0%) | 11 (10.5%) | 80 (76.2%) | 12 (11.4%) | 2 (1.9%) | 105 (7.5%) | |
| Popular breed | Burmese | 1 (2.8%) | 2 (5.6%) | 27 (75.0%) | 6 (16.7%) | 0 (0.0%) | 36 (2.6%) | 1,390 (100.0%) |
| | Mixed-breed | 9 (0.8%) | 61 (5.5%) | 746 (67.1%) | 272 (24.5%) | 24 (2.2%) | 1,112 (80.0%) | |
| | Ragdoll | 0 (0.0%) | 8 (13.3%) | 49 (81.7%) | 3 (5.0%) | 0 (0.0%) | 60 (4.3%) | |
| | Siamese | 0 (0.0%) | 4 (10.8%) | 31 (83.8%) | 2 (5.4%) | 0 (0.0%) | 37 (2.7%) | |
| | Other purebred | 1 (0.7%) | 14 (9.7%) | 101 (69.7%) | 23 (15.9%) | 6 (4.1%) | 145 (10.4%) | |
| Hair length | Shorthaired | 5 (0.6%) | 51 (6.4%) | 546 (68.4%) | 170 (21.3%) | 26 (3.3%) | 798 (57.5%) | 1,388 (99.9%) |
| | Medium-haired | 3 (0.8%) | 20 (5.3%) | 251 (67.1%) | 98 (26.2%) | 2 (0.5%) | 374 (26.9%) | |
| | Longhaired | 3 (1.4%) | 18 (8.3%) | 155 (71.8%) | 38 (17.6%) | 2 (0.9%) | 216 (15.6%) | |
| Sex | I am not sure[2] | 0 (0.0%) | 0 (0.0%) | 3 (100.0%) | 0 (0.0%) | 0 (0.0%) | 3 (0.2%) | 1389 (99.9%) |
| | Female | 7 (1.0%) | 53 (7.4%) | 503 (69.9%) | 138 (19.2%) | 19 (2.6%) | 720 (51.8%) | |
| | Male | 4 (0.6%) | 36 (5.4%) | 448 (67.3%) | 167 (25.1%) | 11 (1.7%) | 666 (47.9%) | |
| Neuter status[3] | I am not sure | 0 (0.0%) | 0 (0.0%) | 3 (100.0%) | 0 (0.0%) | 0 (0.0%) | 3 (0.2%) | 1,386 (99.71%) |
| | Intact | 0 (0.0%) | 0 (0.0%) | 27 (100.0%) | 0 (0.0%) | 0 (0.0%) | 27 (1.9%) | |
| | Neutered | 11 (0.8%) | 89 (6.6%) | 920 (67.8%) | 306 (22.6%) | 30 (2.2%) | 1,356 (97.8%) | |
| Neutering age | I am not sure | 0 (0.0%) | 13 (7.5%) | 118 (67.8%) | 41 (23.6%) | 2 (1.1%) | 174 (12.7%) | 1,368 (98.4%) |
| | 0–<3 months | 3 (0.9%) | 13 (3.8%) | 231 (67.3%) | 84 (24.5%) | 12 (3.5%) | 343 (25.1%) | |
| | 3–<6 months | 4 (0.8%) | 39 (7.6%) | 358 (69.6%) | 105 (20.4%) | 8 (1.6%) | 514 (37.6%) | |
| | 6–<12 months | 1 (0.5%) | 14 (6.7%) | 139 (66.2%) | 51 (24.3%) | 5 (2.4%) | 210 (15.4%) | |
| | 1–<3 years | 3 (3.1%) | 6 (6.1%) | 69 (70.4%) | 18 (18.4%) | 2 (2.0%) | 98 (7.2%) | |
| | ≥3 years | 0 (0.0%) | 3 (10.3%) | 18 (62.1%) | 7 (24.2%) | 1 (3.4%) | 29 (2.1%) | |

[1]: Cats were considered purebred if they were registered with the Australian Cat Federation.

[2]: Sex was regarded as missing if the owners were not sure about the sex of the cat.

[3]: Neuter status was not included as an explanatory variable because all the intact cats had ideal weight.

to conduct the Hosmer-Lemeshow test to examine the quality of the final multinomial model fit [38]. This test was accomplished with the 'ResourceSelection' package [39].

**Reliability of the responses.** The reliability of the responses was estimated by Cohen's kappa with 'psych' package [33] by comparing the responses of two hunting-related questions. The question asked whether the cat hunted, and the reported frequency of hunting was re-categorised as 'hunting', 'not hunting' and 'I am not sure'. The other question asked what prey the cat hunted. If the participants named any type of prey in the response, their cats were considered to be cats that hunted; otherwise, the cats remained in the same 'not hunting' or 'I am not sure' category. Responses that contained missing answers to any of the two questions were excluded.

## Results

### Descriptive results

**Response.** Of the 1,469 questionnaires completed, 1,390 questionnaires included participants' evaluation of the BCS of their cats and were thus integrated into the analysis. More than

half of the participants heard about the survey from social media (691, 53.0%) and 29.6% (n = 386) of the participants obtained the survey from RSPCA Australia-related sources.

**Demographics of the participants.** Of the 1,390 participants, 1,186 (89.0%) were female and approximately half were aged between 25 and 44 years (720, 51.8%). Eighty-nine (6.7%) participants did not live in Australia at the time of completing the questionnaire. For the other 1,242 (93.3%), the three states with the most participants were New South Wales (493, 41.0%), Victoria (264, 22.0%) and Queensland (230, 19.2%). Among Australian residential participants, 824 (68.6%) lived in an urban area and 377 (23.1%) lived in a rural area. More than half of the participants had a bachelor's degree or higher (719, 54.0%) and 199 (15.3%) worked in an animal-related profession. Around half (684, 49.2%) of the participants owned only one cat, 450 (32.4%) owned two cats and 255 (18.4%) owned three or more cats. More participant demographics can be found in S4 Table.

**Demographics of the cats.** The detailed demographics of the 1390 cats are displayed in Table 1. Nearly half of the cats were ≥3 to <11 years old (48.3%) and the majority were mixed-breed (80.0%). There were slightly more female cats (51.8%) than males (47.9%) and nearly all of them were neutered (97.8%). The most common source of the cats was a shelter, pound or charity (669, 48.1%), followed by a breeder (198, 14.2%), a friend or family member (184, 13.2%), street cat (138, 9.9%) and a pet shop (80, 5.8%). Summary statistics of variables related to feeding pattern and activity are listed in S2 Table and S3 Table, respectively.

**BCS results.** The numbers of cats evaluated with 'BCS Owner', 'BCS Figure' and 'BCS Vet' are presented in Fig 1. More than two-thirds (954, 68.6%) of cats were considered to have a BCS Owner of 3, and BCS Figure tended to be judged lower than BCS Owner and BCS Vet. Of the 1,390 cats, the BCS of 571 (41.1%) cats had not been evaluated by a veterinarian in the previous year. The comparisons of the three BCS evaluated are shown in Table 2. Only 126 (9.1%) participants weighed their cats regularly, 318 (22.9%) weighed their cats from time to time, 593 (42.8%) were informed of the cat's weight when visiting a veterinarian, and 349 (25.2%) did not monitor at all.

BCS Owner denotes participant-perceived BCS, as categorised by participants based on descriptions in words in the questionnaire; BCS Figure was determined by participants by selecting one picture of a cat whose shape was the most similar to their cats out of five pictures; BCS Vet was the BCS determined by veterinarians in the previous year and reported by the participants. 'Other' of BCS Figure includes the cats that were not nearby the participants while answering the questionnaire, so their BCS could not be compared to the figure and so a category according to the figure could not be recorded. 'Other' of BCS Vet indicates a cat that had not been evaluated by a veterinarian in the previous year.

Weighted kappa suggested that BCS Owner had better agreement with both BCS Vet and BCS from Cat Owner Club than BCS Figure (Table 2). Nevertheless, 20.8% (n = 44) of the cats evaluated by veterinarians to have a BCS4 were considered to have a BCS of 3 by owners, indicating that owners underestimated their cats' body condition. On the other hand, 22.2% (n = 14) of cats evaluated by veterinarians to have a BCS of 2 were considered to have a BCS of 3 by the owners, indicating that owners were more likely to report ideal body condition scores. However, the prevalence of O&O estimated using BCS Owner was only slightly lower (24.2%) than the prevalence of 25.9%, calculated using BCS Vet. Both the values of weighted kappa with BCS from Cat Owner Club were low, especially the one with BCS Figure, whose 95% confidence interval covered 0, implying a non-effective agreement.

**Descriptive statistics of the attitude towards feline overweight and obesity among the participants.** The descriptive results of the attitude-related questions are presented in Table 3. In general, more participants chose 'disagree' or 'neutral' than other options in these 10 questions. About a quarter of the participants strongly disagreed with two statements: 'I

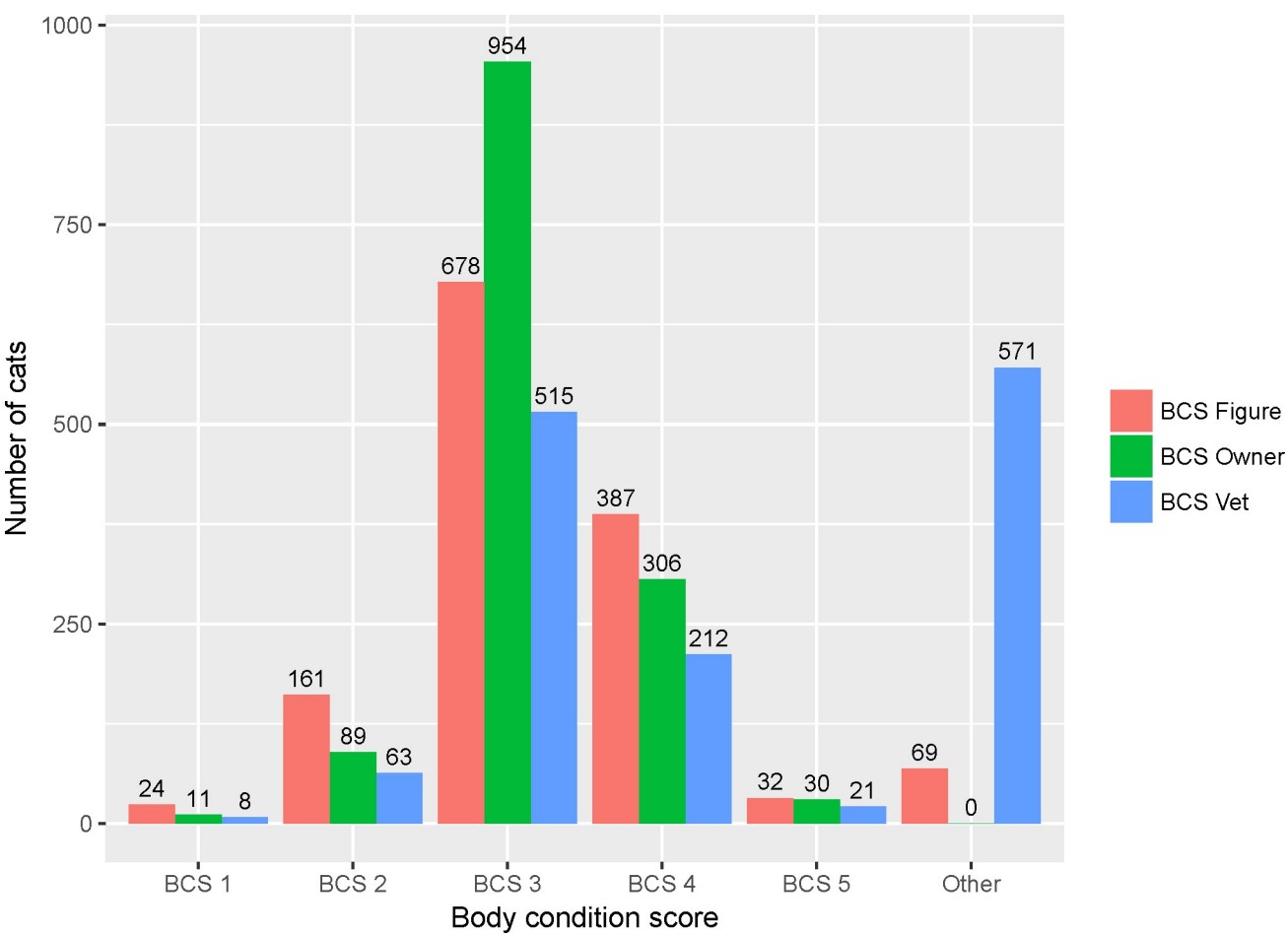

**Fig 1. Body condition scores (BCS; 1 to 5) for 1,390 cats evaluated by different approaches, based on data collected by an Australian-based online survey in 2016.**

think that it's fine for cats to be fat' and 'Being fat says that the cat has a quality life.' However, a quarter of the participants (362, 26.0%) thought chubby cats were cute, and more than one-sixth (252, 18.1%) agreed that fat cats were cute. The same pattern appeared in the statements 'Being chubby doesn't equal unhealthy' and 'Being fat doesn't equal unhealthy'.

## Association between attitude and body condition score

Ten univariable models were fitted for the 10 questions (Table 4). Apart from the model with the answers to the question 'fat cats are cute', all models were statistically significant. In these nine significant models, the odds of being overweight or obese in the cats with the owners who had an approving attitude towards overweight and obesity were higher than in the cats with the owners with a disapproving attitude. Although the odds of being overweight or obese in the cats with the owners with a neutral attitude were all higher than the cats with the owners with a disapproving attitude in the nine overall significant models, only five comparisons were statistically significant. Cats had particularly high odds of overweight and obesity, respectively, if their owner agreed that 'being chubby says that the cat has a quality life' (OR: 3.75, 95% CI: 2.41–5.82) and 'being fat says that the cat has a quality life' (OR: 4.98, 95%CI: 2.79–8.91).

**Table 2. The tabulated body condition score (BCS) evaluations and weighted kappa between different owner-reported BCS based on data collected by an Australian-based online survey in 2016 (n = 1,390).**

| | | BCS Owner[1] | | | | | BCS Figure[2] | | | | |
|---|---|---|---|---|---|---|---|---|---|---|---|
| | | 1 | 2 | 3 | 4 | 5 | 1 | 2 | 3 | 4 | 5 |
| BCS Vet[3] | 1 | 5 (62.5%) | 0 | 3 (37.5%) | 0 | 0 | 3 (42.9%) | 1 (14.3%) | 3 (42.9%) | 0 | 0 |
| | 2 | 5 (7.9%) | 43 (67.2%) | 14 (22.2%) | 1 (1.6%) | 0 | 7 (11.9%) | 23 (40.0%) | 24 (40.7%) | 5 (8.5%) | 0 |
| | 3 | 0 | 15 (2.9%) | 481 (93.4%) | 17 (3.3%) | 2 (0.4%) | 8 (1.7%) | 79 (16.7%) | 317 (67.0%) | 67 (14.2%) | 2 (0.4%) |
| | 4 | 1 (0.5%) | 0 | 44 (20.8%) | 161 (75.9%) | 6 (2.8%) | 0 | 4 (2.0%) | 43 (21.4%) | 140 (69.7%) | 14 (7.0%) |
| | 5 | 0 | 0 | 0 | 4 (19.0%) | 17 (81.0%) | 0 | 0 | 0 | 13 (61.9%) | 8 (38.1%) |
| | Sum | 11 | 58 | 532 | 183 | 25 | 18 | 107 | 387 | 225 | 24 |
| Weighted kappa | BCS Vet | 0.81 (95%CI: 0.77–0.85) | | | | | 0.59 (95%CI: 0.55–0.64) | | | | |
| | BCS from Cat Owner Club[4] | 0.10 (95%CI: 0.04–0.16) | | | | | 0.03 (95%CI: -0.05–0.10) | | | | |

[1]: Owner-perceived BCS of their cats

[2]: BCS that owners determined by selecting a picture of a cat that was most similar to the shape of their cats

[3]: Owner-reported BCS determined by veterinarians in the previous year

[4]: Grouped BCS using the data from the Cat Owner Club website (https://www.catownerclub.com/cat-breeds).

## Risk factors for feline overweight and obesity and underweight

**Multinomial model results.** The univariable results of 27 explanatory variables (including three confounders) that had a *P*-value less than 0.2 out of 47 variables investigated (including five confounders) are presented in Table 5. Among the 47 explanatory variables, two (*home-made cat food* and *leftovers of human food*) had greater than 15% of data missing (i.e., 16.6% and 16.3%, respectively). No collinearity between explanatory variables was detected.

The final model results are presented in Table 6. Eight variables were statistically significantly associated with the outcome variable, and two confounders, education level and animal-related profession, were included in the final model. No tested interaction terms were significant. In the final model, the factors associated with an increased odds of feline O&O

**Table 3. Descriptive results of the questions exploring participants' attitudes towards overweight (i.e., 'chubby' in the statements) and obesity (i.e., 'fat' in the statements) in cats in an Australian-based online survey in 2016 (n = 1,390).**

| Question | Strongly disagree | Disagree | Neutral | Agree | Strongly agree | Total |
|---|---|---|---|---|---|---|
| Chubby cats are cute | 158 (11.4%) | 385 (27.7%) | 443 (31.9%) | 334 (24.0%) | 28 (2.0%) | 1348 (97.0%) |
| Fat cats are cute | 227 (16.3%) | 490 (35.3%) | 379 (27.3%) | 231 (16.6%) | 21 (1.5%) | 1348 (97.0%) |
| Chubby cats usually look happier | 191 (13.7%) | 615 (44.2%) | 442 (31.8%) | 92 (6.6%) | 9 (0.6%) | 1349 (97.1%) |
| Fat cats usually look happier | 269 (19.4%) | 657 (47.3%) | 359 (25.8%) | 63 (4.5%) | 3 (0.2%) | 1351 (97.2%) |
| I think that it's fine for cats to be chubby | 158 (11.4%) | 590 (42.4%) | 379 (27.3%) | 211 (15.2%) | 12 (0.9%) | 1350 (97.1%) |
| I think that it's fine for cats to be fat | 350 (25.2%) | 714 (51.4%) | 223 (16.0%) | 62 (4.5%) | 3 (0.2%) | 1352 (97.3%) |
| Being chubby doesn't equal unhealthy | 127 (9.1%) | 458 (32.9%) | 378 (27.2%) | 355 (25.5%) | 31 (2.2%) | 1349 (97.1%) |
| Being fat doesn't equal unhealthy | 212 (15.3%) | 578 (41.6%) | 332 (23.9%) | 211 (15.2%) | 14 (1.0%) | 1347 (96.9%) |
| Being chubby says that the cat has a quality life | 256 (18.4%) | 675 (48.6%) | 322 (23.2%) | 93 (6.7%) | 6 (0.4%) | 1352 (97.3%) |
| Being fat says that the cat has a quality life | 348 (25.0%) | 711 (51.2%) | 237 (17.1%) | 51 (3.7%) | 3 (0.2%) | 1350 (97.1%) |

**Table 4. The odds ratios for owner-assessed overweight and obesity compared to owner-assessed ideal weight in the cats of participants against participant attitude towards feline overweight (i.e., 'chubby' in the statements) or obesity (i.e., 'fat' in the statements) according to the information collected by an Australian-based online questionnaire in 2016 (n = 1,390).**

| Question | Attitude[1] | OR[2] (95% CI[3]) | P-value | Overall P-value |
|---|---|---|---|---|
| Chubby cats are cute | Approving | 1.66 (1.22–2.26) | 0.001 | 0.005 |
| | Neutral | 1.16 (0.85–1.57) | 0.353 | |
| | Disapproving | 1 | - | |
| Fat cats are cute | Approving | 1.42 (1.02–1.97) | 0.039 | 0.101 |
| | Neutral | 1.21 (0.90–1.62) | 0.216 | |
| | Disapproving | 1 | - | |
| Chubby cats usually look happier | Approving | 2.35 (1.49–3.69) | <0.001 | <0.001 |
| | Neutral | 1.70 (1.29–2.23) | <0.001 | |
| | Disapproving | 1 | - | |
| Fat cats usually look happier | Approving | 2.34 (1.35–4.03) | 0.002 | 0.002 |
| | Neutral | 1.40 (1.05–1.86) | 0.020 | |
| | Disapproving | 1 | - | |
| I think that it's fine for cats to be chubby | Approving | 2.06 (1.47–2.89) | <0.001 | <0.001 |
| | Neutral | 1.52 (1.13–2.04) | 0.005 | |
| | Disapproving | 1 | - | |
| I think that it's fine for cats to be fat | Approving | 2.10 (1.23–3.61) | 0.007 | 0.026 |
| | Neutral | 1.16 (0.83–1.63) | 0.388 | |
| | Disapproving | 1 | - | |
| Being chubby doesn't equal unhealthy | Approving | 1.83 (1.36–2.47) | <0.001 | <0.001 |
| | Neutral | 1.13 (0.82–1.55) | 0.455 | |
| | Disapproving | 1 | - | |
| Being fat doesn't equal unhealthy | Approving | 1.54 (1.10–2.16) | 0.012 | 0.041 |
| | Neutral | 1.19 (0.88–1.62) | 0.259 | |
| | Disapproving | 1 | - | |
| Being chubby says that the cat has a quality life | Approving | 3.75 (2.41–5.82) | <0.001 | <0.001 |
| | Neutral | 1.42 (1.05–1.91) | 0.021 | |
| | Disapproving | 1 | - | |
| Being fat says that the cat has a quality life | Approving | 4.98 (2.79–8.91) | <0.001 | <0.001 |
| | Neutral | 1.20 (0.86–1.67) | 0.283 | |
| | Disapproving | 1 | - | |

[1]. Each question provided five answer options (strongly disagree, disagree, neutral, agree and strongly agree). Answers for each question were grouped as having an approving attitude (towards overweight or obesity), neutral attitude or disapproving attitude.

[2]: Odds ratio

[3]: Confidence interval

included being middle-aged, being mixed-breed, being fed twice daily, being fed more than four times a day or *ad libitum*, dry food being the major part of the diet, participants who determined the food quantity without considering the amount that cats ate, begging for food by cats, staying less often outdoors outside the property of the owner and apartment and townhouse dwelling. The factors associated with an increased odds of underweight in cats included being aged 11 years or older, being fed four or more times a day, and townhouse dwelling. The final model showed a good fit with the *P*-values of the approximated Hosmer-Lemeshow test >0.05 (i.e., 0.888 and 0.876 for models with binomial outcomes, underweight versus ideal weight and overweight versus ideal weight, respectively).

**Table 5. Univariable results of the multinomial logistic regression model to evaluate risk factors for owner-assessed underweight and overweight and obesity (O&O) in cats with a *P*-value less than 0.2, based on data from 1,390 cats collected by an Australian-based online questionnaire in 2016.**

| Variable | Category | Underweight versus ideal weight | | O&O[3] versus ideal weight | | Overall *P*-value |
|---|---|---|---|---|---|---|
| | | OR[1] (95% CI[2]) | *P*-value | OR (95% CI[1]) | *P*-value | |
| **Cat demographics** | | | | | | |
| Age | <1 year | 1 | - | 1 | - | <0.001* |
| | ≥1 to <3 years | 0.56 (0.22–1.46) | 0.239 | 2.15 (1.26–3.66) | 0.005* | |
| | ≥3 to <11 years | 1.12 (0.54–2.35) | 0.761 | 3.65 (2.25–5.92) | <0.001* | |
| | ≥15 years | 6.89 (3.37–14.1) | <0.001* | 2.90 (1.65–5.12) | <0.001* | |
| Breed | Mixed | 1 | - | 1 | - | <0.001* |
| | Pedigree | 1.47 (0.75–2.88) | 0.268 | 0.44 (0.25–0.79) | 0.006* | |
| | Purebred | 1.58 (0.92–2.72) | 0.096 | 0.51 (0.33–0.80) | 0.003* | |
| Hair length | Shorthaired | 1 | - | 1 | - | 0.139 |
| | Medium-haired | 0.89 (0.54–1.48) | 0.664 | 1.11 (0.84–1.47) | 0.471 | |
| | Longhaired | 1.32 (0.78–2.25) | 0.305 | 0.72 (0.49–1.06) | 0.092 | |
| Sex | Female | 1 | - | 1 | - | 0.040* |
| | Male | 0.75 (0.49–1.14) | 0.176 | 1.27 (0.99–1.63) | 0.058 | |
| **Feeding frequency and Food type** | | | | | | |
| Daily feeding frequency | 1/day | 1 | - | 1 | - | 0.002* |
| | 2/day | 0.88 (0.45–1.70) | 0.695 | 1.61 (1.05–2.47) | 0.028* | |
| | 3/day | 1.59 (0.72–3.49) | 0.252 | 1.38 (0.80–2.40) | 0.245 | |
| | ≥4/day | 4.11 (1.61–10.48) | 0.003* | 2.46 (1.17–5.19) | 0.018* | |
| | Ad libitum | 1.91 (0.86–4.23) | 0.110 | 1.86 (1.08–3.21) | 0.026* | |
| Dry food | Not part of the diet | 1 | - | 1 | - | <0.001* |
| | Minor diet | 0.29 (0.12–0.72) | 0.007* | 0.43 (0.20–0.94) | 0.035* | |
| | Major diet | 0.18 (0.08–0.44) | <0.001* | 0.63 (0.30–1.36) | 0.245 | |
| Wet food apart from cans | Not part of the diet | 1 | - | 1 | - | 0.156 |
| | Minor diet | 0.83 (0.49–1.42) | 0.501 | 0.93 (0.68–1.27) | 0.633 | |
| | Major diet | 1.20 (0.72–2.02) | 0.486 | 0.71 (0.50–1.01) | 0.057 | |
| Leftover of human food | Not part of the diet | 1 | - | 1 | - | 0.036* |
| | Sometimes | 1.28 (0.74–2.20) | 0.379 | 0.73 (0.55–0.97) | 0.033* | |
| | Often | 1.99 (1.06–3.75) | 0.033* | 0.62 (0.41–0.94) | 0.023* | |
| | Always | 1.29 (0.55–3.04) | 0.564 | 0.72 (0.43–1.19) | 0.197 | |
| **Methods used to determine the quantity of food** | | | | | | |
| No specific rules | No | 1 | - | 1 | - | 0.035* |
| | Yes | 0.48 (0.26–0.89) | 0.020* | 1.03 (0.77–1.39) | 0.825 | |
| Advice from veterinarians | No | 1 | - | 1 | - | 0.116 |
| | Yes | 1.43 (0.91–2.23) | 0.118 | 1.26 (0.96–1.67) | 0.101 | |
| Advice from the package | No | 1 | - | 1 | - | 0.001* |
| | Yes | 0.41 (0.23–0.73) | 0.003* | 1.18 (0.90–1.54) | 0.238 | |
| According to the amount my cat eats | No | 1 | - | 1 | - | <0.001* |
| | Yes | 2.15 (1.41–3.25) | <0.001* | 0.53 (0.40–0.72) | <0.001* | |
| Providing more than my cat needs | No | 1 | - | 1 | - | 0.021* |
| | Yes | 1.97 (0.80–4.86) | 0.140 | 2.18 (1.24–3.83) | 0.007* | |
| **Other feeding-related factors** | | | | | | |
| Food begging behaviours | Never | 1 | - | 1 | - | <0.001* |
| | Sometimes | 1.32 (0.78–2.23) | 0.296 | 1.46 (1.04–2.05) | 0.028* | |
| | Often | 1.67 (0.82–3.41) | 0.157 | 3.09 (2.04–4.70) | <0.001* | |
| | Always | 2.69 (1.19–6.08) | 0.017* | 4.45 (2.69–7.35) | <0.001* | |

*(Continued)*

**Table 5.** (Continued)

| Variable | Category | Underweight versus ideal weight | | O&O[3] versus ideal weight | | Overall P-value |
|---|---|---|---|---|---|---|
| | | OR[1] (95% CI[2]) | P-value | OR (95% CI[1]) | P-value | |
| Owner giving in to begging | Never | 1 | - | 1 | - | <0.001* |
| | Sometimes | 1.47 (0.85–2.54) | 0.163 | 1.41 (1.06–1.89) | 0.020* | |
| | Often | 3.36 (1.75–6.46) | <0.001* | 1.99 (1.32–3.00) | 0.001* | |
| | Always | 3.70 (1.84–7.45) | <0.001* | 0.79 (0.43–1.45) | 0.448 | |
| Food sources other than the owner | No | 1 | - | 1 | - | 0.186 |
| | Yes | 0.69 (0.24–1.95) | 0.483 | 1.47 (0.91–2.38) | 0.115 | |
| **Activities of the cat** | | | | | | |
| Staying outdoors but still inside the property | Not often | 1 | - | 1 | - | 0.138 |
| | Often | 0.76 (0.48–1.21) | 0.244 | 0.78 (0.59–1.03) | 0.081 | |
| Staying outdoors but outside of the property | Not often | 1 | - | 1 | - | 0.048* |
| | Often | 0.69 (0.24–1.94) | 0.479 | 0.44 (0.22–0.91) | 0.026* | |
| Hunting frequency | Never | 1 | - | 1 | - | 0.008* |
| | Sometimes | 0.70 (0.44–1.12) | 0.140 | 0.90 (0.67–1.20) | 0.467 | |
| | Often | 0.32 (0.12–0.81) | 0.017* | 0.53 (0.33–0.86) | 0.009* | |
| Prey type | Large animal[4] | 1 | - | 1 | - | 0.091 |
| | Not hunting | 1.59 (0.79–3.20) | 0.197 | 1.39 (0.91–2.12) | 0.132 | |
| | Small animals[5] | 0.91 (0.43–1.92) | 0.801 | 1.22 (0.79–1.88) | 0.372 | |
| Owner playing with the cat | Not often | 1 | - | 1 | - | 0.078 |
| | Often | 0.65 (0.42–1.01) | 0.054 | 0.73 (0.56–0.96) | 0.025* | |
| | Always | 0.58 (0.28–1.23) | 0.156 | 0.78 (0.51–1.18) | 0.235 | |
| **Owner demographics and home-related figures** | | | | | | |
| Education level | Secondary school qualification | 1 | - | 1 | - | 0.123 |
| | TAFE[6]/ VET[7] qualification[8] | 2 (0.97–4.11) | 0.061 | 1.21 (0.81–1.80) | 0.357 | |
| | Bachelors degree[8] | 1.67 (0.83–3.37) | 0.154 | 1.35 (0.93–1.95) | 0.111 | |
| | Masters degree[8] | 1.88 (0.82–4.35) | 0.137 | 1.20 (0.74–1.92) | 0.461 | |
| | Doctoral degree[8] | 1.74 (0.46–6.61) | 0.416 | 1.47 (0.70–3.08) | 0.306 | |
| | Other | 10.51 (2.95–37.51) | <0.001* | 2.67 (0.89–8.03) | 0.081 | |
| Being a veterinarian | No | 1 | - | 1 | - | <0.001* |
| | Yes | 0.17 (0.02–1.23) | 0.079 | 1.94 (1.24–3.03) | 0.004* | |
| Animal-related profession | No | 1 | - | 1 | - | 0.073 |
| | Yes | 0.54 (0.26–1.14) | 0.105 | 0.54 (0.26–1.14) | 0.105 | |
| Household type | Family | 1 | - | 1 | - | 0.162 |
| | Shared household | 1.34 (0.66–2.72) | 0.410 | 0.99 (0.63–1.56) | 0.970 | |
| | Single person | 1.85 (1.13–3.01) | 0.014* | 1.22 (0.88–1.68) | 0.235 | |
| Dwelling type | Apartment | 1 | - | 1 | - | 0.002* |
| | Townhouse | 1.93 (0.95–3.92) | 0.069 | 1.19 (0.77–1.84) | 0.426 | |
| | House | 0.94 (0.54–1.66) | 0.835 | 0.66 (0.48–0.90) | 0.009* | |

[1]: Odd ratio

[2]: Confidence interval

[3]: Overweight and obesity

[4]: Birds and large mammals

[5]: Small mammals, insects, lizards and frogs

[6]: Technical and further education

[7]: Vocational education and training

[8]: Or equivalent

*: P-value < 0.05

**Table 6. Final multivariable multinomial model for risk factors for owner-assessed underweight and overweight and obesity (O&O) in cats based on the model of 1,390 cats whose information was collected by an Australian-based online questionnaire in 2016.**

| Variable | Category | Underweight versus ideal weight | | O&O[3] versus ideal weight | | OverallP-value |
|---|---|---|---|---|---|---|
| | | OR[1] (95% CI[2]) | P-value | OR (95% CI) | P-value | |
| Age of the cat | <1 year | 1.85 (0.59–5.84) | 0.291 | 0.42 (0.23–0.77) | 0.005* | <0.001* |
| | ≥1 to <3 years | 1 | - | 1 | - | |
| | ≥3 to <11 years | 2.51 (0.98–6.45) | 0.056 | 1.64 (1.13–2.38) | 0.009* | |
| | ≥11 years | 14.09 (5.53–35.94) | <0.001* | 1.49 (0.88–2.51) | 0.137 | |
| Breed of the cat | Mixed | 1 | - | 1 | - | 0.001* |
| | Pedigree | 1.97 (0.87–4.44) | 0.103 | 0.60 (0.31–1.17) | 0.131 | |
| | Purebred | 0.99 (0.48–2.04) | 0.987 | 0.40 (0.24–0.68) | 0.001* | |
| Daily feeding frequency | 1/day | 1 | - | 1 | - | 0.006* |
| | 2/day | 0.81 (0.35–1.89) | 0.631 | 1.82 (1.09–3.04) | 0.021* | |
| | 3/day | 0.95 (0.33–2.69) | 0.920 | 1.44 (0.75–2.75) | 0.273 | |
| | ≥4/day | 4.07 (1.17–14.11) | 0.027* | 3.10 (1.17–8.24) | 0.023 | |
| | Ad libitum | 2.15 (0.80–5.76) | 0.130 | 2.34 (1.23–4.48) | 0.010* | |
| Dry food | Not part of the diet | 4.87 (1.41–16.75) | 0.012* | 1.04 (0.40–2.69) | 0.942 | 0.012* |
| | Minor diet | 1.67 (0.97–2.88) | 0.064 | 0.71 (0.51–1.00) | 0.047* | |
| | Major diet | 1 | - | 1 | - | |
| Food quantity provided: according to the amount my cat eats | No | 1 | - | 1 | - | <0.001* |
| | Yes | 1.95 (1.15–3.31) | 0.013* | 0.60 (0.43–0.85) | 0.004* | |
| Food begging behaviours | Never | 1 | - | 1 | - | <0.001* |
| | Sometimes | 1.04 (0.54–2.00) | 0.901 | 1.61 (1.08–2.39) | 0.019* | |
| | Often | 1.40 (0.58–3.40) | 0.453 | 3.41 (2.07–5.63) | <0.001* | |
| | Always | 2.26 (0.78–6.58) | 0.134 | 5.19 (2.83–9.51) | <0.001* | |
| Staying outdoors but outside of the property | Less often | 1 | - | 1 | - | 0.010* |
| | Often | 0.67 (0.21–2.08) | 0.483 | 0.34 (0.16–0.73) | 0.006* | |
| Education level of the owner | Secondary school qualification | 1 | - | 1 | - | 0.428 |
| | TAFE[4]/ VET[5] qualification[6] | 1.87 (0.77–4.51) | 0.164 | 0.98 (0.61–1.57) | 0.936 | |
| | Bachelors degree[6] | 1.88 (0.80–4.40) | 0.148 | 1.17 (0.76–1.82) | 0.471 | |
| | Masters degree[6] | 1.64 (0.58–4.61) | 0.349 | 0.97 (0.55–1.69) | 0.902 | |
| | Doctoral degree[6] | 1.92 (0.41–8.99) | 0.409 | 1.26 (0.54–2.91) | 0.592 | |

*(Continued)*

**Table 6.** (Continued)

| Variable | Category | Underweight versus ideal weight | | O&O[3] versus ideal weight | | Overall P-value |
|---|---|---|---|---|---|---|
| | | OR[1] (95% CI[2]) | P-value | OR (95% CI) | P-value | |
| | Other | 12.54 (2.52–62.40) | 0.002* | 1.46 (0.38–5.61) | 0.579 | |
| Animal-related profession | No | 1 | - | 1 | - | 0.950 |
| | Yes | 0.93 (0.41–2.12) | 0.856 | 1.05 (0.7–1.58) | 0.813 | |
| Dwelling type | Apartment | 1.47 (0.76–2.86) | 0.257 | 1.43 (0.99–2.06) | 0.055 | 0.007* |
| | Townhouse | 2.44 (1.18–5.02) | 0.016* | 1.92 (1.25–2.93) | 0.003* | |
| | House | 1 | - | 1 | - | |

[1]: Odd ratio

[2]: Confidence interval

[3]: Overweight and obesity

[4]: Technical and further education

[5]: Vocational education and training

[6]: Or equivalent

*: P-value < 0.05

### The reliability of the responses

Cohen's kappa for two hunting-related questions was 0.83 (95% CI: 0.80–0.86), which is considered very good agreement [40].

## Discussion

The current study examined the risk factors for underweight and O&O in cats and, for the first time, reports the association between feline O&O and the owners' attitudes towards O&O in cats.

### Demographics of the participants

Compared with the Australian Bureau of Statistics 2016 Census Data [41], our study population had higher proportions of people aged between 25 and 44 years of age (51.8% vs. 35.3%), female (85.3% vs. 50.7%), those with a tertiary qualification (54.6% vs. 25.1%), residents in New South Wales (41.0% vs. 32.0%) and those living in an apartment dwelling type (20.3% vs. 13.1%). It was unsurprising that female owners dominated the survey. A survey in 2016 showed that 76% of cat owners in Australia were female [42], and females are more likely to show an empathetic attitude towards animals and be interested in animal welfare [43–45]. Thus, cat husbandry attributes and the attitude towards O&O reported in the current study reflect the opinions of Australian, educated women rather than general Australian cat owners.

### Demographics of the cats

There was a slightly higher proportion of female cats than males in the current study (*P* = 0.323); this trend has been observed in several Australian-based feline studies [14, 20, 46, 47]. In the current study, most cats were neutered (97.8%), and the percentage of neutering was higher than the overall Australian population (89%) [42]. Twenty percent of cats in the current study population was purebred or pedigree, slightly lower than two other Australian-

wide statistics at 23.4% [46] and 24% [42] and New South Wales Companion Animals Register (22.4%) [48]. Although Australian council registration rates and cat health insurance rates were estimated at 72% and 19% in 2016 [42], respectively, they were only 63.1% and 11.3% in the current study.

## The body condition score evaluation

There were three candidates for the outcome variable for BCS evaluation, namely, BCS Owner, BCS Figure and BCS Vet. We could not use BCS Vet due to a large number of missing values. It has been consistently shown that owners often underestimate cat BCS when directly asked the BCS of their cats, and the underestimation is a risk factor for feline O&O [8, 10, 11]. However, in contrast to what we had presumed based on the literature, BCS Owner better reflected BCS Vet than BCS Figure. That said, we acknowledge that BCS Vet might also be biased as it was reported by the owner and not obtained directly from the veterinarian (i.e., may have been subject to recall bias). It was noted that, compared with the distribution of BCS Owner, the distribution of BCS Figure was more dispersed. A possible explanation is that the owners might answer the BCS Owner question by referring to the BCS evaluated by veterinarians. Also, providing only images without further description might likely be insufficient for owners to judge the BCS of their cats [9]. Interestingly, Eastland-Jones, German [49] concluded that the accuracy of owner-perceived BCS in dogs was not improved by consulting a 5-point BCS chart.

## Owners attitudes towards feline overweight and obesity

Our study shows that more participants held a disapproving attitude towards feline O&O (39.1–76.6% in the ten questions) than those who had a neutral (17.1–31.9%) or approving attitude (3.9–27.7%). Only a small proportion of the participants had a strong positive attitude towards feline O&O (0.2%–2.2%). Also, the participants had a more disapproving attitude towards feline obesity than overweight. Two sets of questions, 'Chubby/fat cats are cute' and 'Being chubby/fat doesn't equal unhealthy' received particularly lower levels of disagreement than other questions. In contrast to overweight or obese human individuals who are often stigmatised and [23, 50], people seem to have less prejudice against overweight or obese cats. So, it is anticipated that overweight and obese cats are not perceived negatively in the same ways as occurs for overweight and obese humans, such as lacking in self-discipline or being less competent. Indeed, it is recognised that many people even consider chubbiness and fatness as representing cuteness in cats [24, 25]. Adaption from Eibl-Eibesfeldt and Klinghammer [51] and Lorenz [52], Genosko [53] proposed seven physical traits and one behavioural trait that typify feline cuteness, of which short, stubby limbs with pudgy paws and hands, rounded, fat body shape, soft, elastic body surfaces and clumsiness are often manifested in overweight and obese cats. Moreover, the fat-cuteness perception is also likely to be reinforced by frequent exposure to cartoons, images and videos of plump cats (and potentially other species). For the second set of questions, the results did not meet our expectation that participants would consider being chubby/fat as unhealthy. Even if the respondents were not familiar with O&O-associated health conditions in cats, O&O in humans has been shown to be associated with various health conditions [54–63]. It is possible that many owners do not connect feline O&O with ill-health. Our results reveal a lack of knowledge of the negative impacts of O&O on feline health among cat owners. The need for owner education in this regard should be emphasised.

Interestingly, while cats whose owners had a positive attitude toward feline O&O had significantly higher odds of being overweight or obese than those with a disapproving attitude in all the models, not all comparisons between the owners with a neutral attitude and those with a disapproving attitude were significant. It is possible that the questions revealing a statistically

significant difference of cat BCS between the owners with a neutral attitude and those with a disapproving attitude are more sensitive to, and better predict, a change in the likelihood of feline O&O. Applying the same logic, the questions stating that both chubby/fat cats usually look happier appear to be good predictors of the likelihood of feline O&O. Having a neutral attitude towards the two statements that 'it is fine to be chubby' and if 'being chubby says that the cat has a quality life' were also risk factors for O&O in respondents' cats. This might be explained by the finding that a greater proportion of respondents agreed that 'being fat/obese is undesirable' than 'being chubby/overweight is undesirable'. So, the questions about attitudes towards chubbiness are likely to be more sensitive to reflecting an indifferent attitude towards O&O among cat owners.

The current results suggest that identifying the attitudes towards O&O among cat owners may be very important and will help to foreshadow the possibility of O&O in their cats. With more evidence revealing the negative impact of O&O on feline health [21, 64, 65], there is an apparent need to boost owner knowledge of feline O&O and its harmful consequences.

## Risk factor analyses

In the current study, the owner-reported BCS was classified into three categories: underweight, ideal weight, and overweight and obese, as we also applied in a previous study [20]. This approach is preferable to that applied in many studies investigating risk factors in which O&O was compared to ideal-weight-and-underweight combined because it avoids the assumption that the mechanisms for change from ideal weight to underweight are similar to those for weight loss in overweight and obese cats. This is reinforced by our finding reported in Teng, McGreevy [20] that the risk factors for O&O are often not protective factors for underweight in cats. As most studies investigating risk factors have not separated underweight and ideal-weight cats or have simply excluded underweight cats from their analyses, the literature about the risk factors for underweight in cats is scarce.

**Age.**  Cat body condition is known to be associated with age. Concurring with previous studies, our results showed the highest odds of O&O in middle-aged cats [6, 11, 13, 18–20, 64] and underweight in elderly cats [15, 20]. As with dogs and humans, there is a tendency for O&O to develop with age [66] because of a reduced energy requirement in cats as they age. However, a cat's ability to digest fat (and potentially protein) also decreases with age, resulting in reduced energy absorption [67]. This outcome is particularly profound in cats over 12 years old and can be one of the reasons for the high frequency of underweight in old cats [67]. Some diseases common among old cats, such as hyperthyroidism [68] and diabetes mellitus [69], result in weight loss. Moreover, inappetence may reflect reduced senses of taste and smell in old cats, or the pain caused by dental and periodontal diseases [66].

**Breed.**  In the current study, mixed-breed cats had significantly higher odds of O&O than purebred cats but not pedigree cats. Mixed-breed has been shown to be associated with feline O&O in several studies [14, 19, 20, 64], although, in these studies, purebred cats were not separated into pedigree and purebred. As pedigree animals are more likely to be raised and cared for not purely for companion purposes, it is possible these cats are treated by their owners differently from purebred cats, resulting in the different odds of O&O in the current study. Corbee [70] reported a 45.5% prevalence of O&O in pedigree show cats, which is higher than most studies investigating the prevalence of feline O&O apart from Russell, Sabin [15]. Although their previous studies have also found a weak association between feline O&O with breed status [11, 17, 71], they all acknowledged that their sample sizes for purebred cats may have lacked power for meaningful comparison.

**Daily feeding frequency.**   In the current study population, most cats (61.0%) were fed twice a day. Compared to this group, cats fed once a day had lower odds of being overweight or obese, and cats being fed at least four times a day or *ad libitum* had higher odds. Relatively frequent daily feeding was associated with higher odds of feline O&O, although being fed three times a day did not show significantly higher odds than the odds of O&O when fed once a day. While several studies have explored the association of feeding frequency with feline O&O [8, 10, 14, 15, 17, 22, 72, 73], only Courcier, O'Higgins [17] reported significant results, in that feeding two to three times a day, but not once a day, was associated with higher odds of O&O than feeding *ad libitum*. In studies that have specifically explored associations between feeding *ad libitum* and feline O&O, inconsistent results have been found. Cave, Allan [10] reported a non-significant association, whereas Russell, Sabin [15] found a higher odds of O&O in cats fed *ad libitum* of canned food but not dry food. It is possible that feeding frequency may be ineffective as a causal factor of feline O&O because, to estimate the total amount of food eaten by cats, other factors such as the amount of food per meal and cats' feeding behaviours must also be considered. Our results also showed that feeding four or more times a day occurred more in underweight cats. The frequent feeding of these cats is likely to be an owner's response to the underweight status instead of the cause of underweight.

**Food type and feeding quantity.**   We examined many types of food as risk factors for O&O and underweight in cats but found that dry food was the only risk factor for O&O. Dry rations have been thought to increase the risk of O&O due to their increased energy density. Although most studies that examined the associations of food types with feline O&O have reported non-significant results, two prospective studies, albeit from the same study population, have reported dry food as a risk factor for feline O&O [9, 22]. Among owners who answer the question of whether treats/snacks were fed to the cats, 57.3% more than half of the cats in the current study population were given treats/snacks (S2 Table). Although feeding treats and/ or table scraps have been reported to relate to feline O&O [6, 9, 15], this association did not emerge in the current study. It is worth noting here that, in contrast to the current study, two of the studies that reported this association did not account for other potential risk factors associated with O&O in the analysis [6, 9, 15].

The amount of food fed to the cats as guided by their apparent appetite was shown to be associated with a lower BCS in the current study. Similarly, in another study, cats with owners who followed the instruction from pet-food companies were reported to have higher odds of O&O than those whose owners determined the amount of food based on their cat's appetite [11]. However, although this practice seems to prevent O&O, it might also increase the odds of underweight in cats, as shown by our results.

**Begging for food.**   To the authors' knowledge, this is the first study to investigate the feline behaviour of begging for food as a potential risk factor for feline O&O. Cats that begged often or always had more than three and five times the odds of being overweight or obese, respectively, than cats that never begged. It appears that, although the owners govern the provision of extra food to their cats, the behaviour of begging is persuasive enough to be a risk factor. Furthermore, besides sometimes using food as a reward, cat owners may sometimes misinterpret attention-seeking behaviours as begging for food and consequently over-feed their cats. This owner response can thus reinforce the behaviour of begging for food among cats [74]. Interestingly, the frequency of the behaviour of begging for food fitted the model better than the frequency of owners giving-in to cats' begging. Therefore, it is possible that the participants underestimated the frequency with which they surrendered to begging cats. Owners should be advised that most attempts to extinguish positively reinforced behaviour are met with a so-called extinction burst whereby the behaviour becomes more frequent before it disappears [75]. Knowing this proximate outcome helps to fortify owners' resolve to ignore escalated

begging. In the current study, begging frequency was related to only O&O but not underweight, indicating that begging is associated with only excessive energy intake but not with the cat's current energy requirement.

**Stay outdoors outside the property.**   In the current study, cats 'often spending time outside the owner's property' had more than three times lower odds of O&O than the cats 'staying less often outside the owner's property'. Although many previous studies have examined the associations between feline O&O and outdoor access [11, 17, 20, 22, 76] or time spent outdoors [9, 14, 15], only Rowe, Browne [22] and Teng, McGreevy [20] reported significant (negative) associations. The outdoor environment not only extends home ranges of cats with outdoor access [77, 78], it also offers them more stimuli and more opportunities to encounter prey. Both these interactions and roaming increase cat energy expenditure and decrease the risk of O&O. However, roaming outdoors also increases the chance of instances that may compromise cat health and welfare such as infectious disease, road traffic accidents and attacks by dogs [74]. Furthermore, predation of wildlife by cats is a severe conservation issue worldwide [79–81]. Possible solutions for cat owners to increase cat physical activity without being released outdoors include a generously sized backyard enclosure and indoor enrichment.

**Dwelling type.**   As confinement seems to be associated with O&O in cats, it is not surprising that cats living in an apartment or townhouse have higher odds of being overweight or obese. Scarlett and Donoghue [13] also reported apartment dwelling as a risk factor for feline O&O. However, cats living in a townhouse were shown to have higher odds of being both underweight and overweight or obese than cats living in a house. The results seem to contradict each other and warrant further investigation. Instead of reflecting any effect of dwelling type, these results might reflect attributes of the cat owners living in different dwelling types that were not measured in this study.

## Limitations

The current study has some limitations in addition to the overrepresentation of female owners and the recall bias of the information about BCS Vet. Firstly, we did not ask whether the participants were the primary caretaker of the cat. This might increase the chance of acquiring inaccurate information if some participants were not the primary caretakers. Secondly, the owner-reported BCS may be subject to misclassification bias. Although we made several attempts to reduce misclassification bias, some were unavoidable in the current study. As discussed above, owners often underestimate the BCS of their cats [8–10]. This manifested in the current results in that the owner-perceived BCS of their cat (i.e., BCS Owner) was idealised (i.e., reported towards BCS 3). This is to be expected because underweight or overweight cats may be considered socially undesirable. The misclassification bias could be non-differential, i.e. it may not depend on other variables. If this is the case, the bias would be towards the null, i.e. the magnitude of parameter estimates would be lower than their true values. Non-differential misclassification would have been a greater concern for the current study had we not identified any effects. However, given that we did find many significant associations, we believe that any effect of non-differential misclassification would not be severe. The differential misclassification bias is also possible in this study, i.e. the classification error is not independent but rather depends on other variables. For example, it is likely that owners who tended to report ideal BCS might also have under-reported inattentive owner practices such as providing more food than the cat needs. Differential misclassification can either exaggerate or underestimate an effect, so the findings of the current study should be interpreted with caution. Lastly, the position of variables in logistic regression cannot imply the direction of causality. For example, in our results, frequent feeding is likely to be an owner's response to their cat's underweight instead of the cause of underweight.

## Conclusions

The current study reveals various factors that are associated with underweight or O&O in cats. It shows, for the first time, that begging for food as a risk factor for feline O&O. Therefore, we suggest this behaviour should be considered and addressed when managing the weight of cats. The results of the current study support the hypothesis that the attitude towards feline O&O among owners is related to the BCS of their cats. As certain attitudes may lead to high-risk behaviours, shifting the approving and neutral attitudes towards feline O&O may potentially reduce the frequency of O&O and associated O&O-related disorders in cats. This could be achieved by identifying the attitude of the owners and equipping them with knowledge of the adverse effects of O&O and reconditioning learned behaviours in both cats and owners. The results of the current study may help to reduce the currently high prevalence of feline O&O.

## Supporting information

**S1 File. The online survey questionnaire.**
(PDF)

**S2 File. The advertisement of the survey for the current study.**
(PDF)

**S1 Table. Intrinsic and extrinsic risk factors investigated for association with feline overweight and obesity (O&O) and reported in the literature.**
(DOCX)

**S2 Table. Contingency tables of potential feeding-related risk factors for feline underweight or overweight and obesity with different body condition scores (BCS 1 to 5) evaluated by 1,390 cat owners, based on data collected by an Australian-based online survey in 2016.**
(DOCX)

**S3 Table. Contingency tables of potential activity-related risk factors for feline underweight or overweight and obesity with different body condition scores (BCS 1 to 5) evaluated by 1,390 cat owners, based on data collected by an Australian-based online survey in 2016.**
(DOCX)

**S4 Table. Contingency tables of participant demographic risk factors for feline underweight or overweight and obesity and potential confounders (i.e., gender, age range, education level, being a veterinarian and animal-related profession) with different body condition scores (BCS 1 to 5) evaluated by 1,390 cat owners, based on data collected by an Australian-based online survey in 2016.**
(DOCX)

## Acknowledgments

The authors thank RSPCA Australia for supporting this study and the participants of the survey.

## Author Contributions

**Conceptualization:** Kendy T. Teng.

**Data curation:** Kendy T. Teng.

**Formal analysis:** Kendy T. Teng.

**Funding acquisition:** Kendy T. Teng, Navneet K. Dhand.

**Investigation:** Kendy T. Teng.

**Methodology:** Kendy T. Teng, Paul D. McGreevy, Jenny-Ann L. M. L. Toribio, Navneet K. Dhand.

**Project administration:** Kendy T. Teng, Navneet K. Dhand.

**Resources:** Kendy T. Teng, Navneet K. Dhand.

**Software:** Kendy T. Teng.

**Supervision:** Paul D. McGreevy, Jenny-Ann L. M. L. Toribio, Navneet K. Dhand.

**Validation:** Kendy T. Teng, Paul D. McGreevy, Jenny-Ann L. M. L. Toribio, Navneet K. Dhand.

**Visualization:** Kendy T. Teng.

**Writing – original draft:** Kendy T. Teng.

**Writing – review & editing:** Kendy T. Teng, Paul D. McGreevy, Jenny-Ann L. M. L. Toribio, Navneet K. Dhand.

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
