## [Decision Letter · Decision Letter 0]

9 Dec 2019

PONE-D-19-30666

Positive attitudes towards feline obesity are strongly associated with ownership of obese cats

PLOS ONE

Dear Dr Teng,

Thank you for submitting your manuscript to PLOS ONE. After careful consideration, we feel that it has merit but does not fully meet PLOS ONE’s publication criteria as it currently stands. Therefore, we invite you to submit a revised version of the manuscript that addresses the points raised during the review process, and which you find detailed below.

We would appreciate receiving your revised manuscript by Jan 23 2020 11:59PM. To enhance the reproducibility of your results, we recommend that if applicable you deposit your laboratory protocols in protocols.io, where a protocol can be assigned its own identifier (DOI) such that it can be cited independently in the future. For instructions see: http://journals.plos.org/plosone/s/submission-guidelines#loc-laboratory-protocols

We look forward to receiving your revised manuscript.

Kind regards,

I Anna S Olsson, Ph.D.

Academic Editor

PLOS ONE

Journal Requirements:

Please ensure that your manuscript meets PLOS ONE's style requirements, including those for file naming. The PLOS ONE style templates can be found at http://www.plosone.org/attachments/PLOSOne_formatting_sample_main_body.pdf and http://www.plosone.org/attachments/PLOSOne_formatting_sample_title_authors_affiliations.pdf We note that you have stated that you will provide repository information for your data at acceptance. Should your manuscript be accepted for publication, we will hold it until you provide the relevant accession numbers or DOIs necessary to access your data. If you wish to make changes to your Data Availability statement, please describe these changes in your cover letter and we will update your Data Availability statement to reflect the information you provide.

Reviewers' comments:

Reviewer's Responses to Questions

**Comments to the Author**

1. Is the manuscript technically sound, and do the data support the conclusions?

Reviewer #1: Partly

Reviewer #2: Yes

2. Has the statistical analysis been performed appropriately and rigorously? 

Reviewer #1: Yes

Reviewer #2: I Don't Know

3. Have the authors made all data underlying the findings in their manuscript fully available?

Reviewer #1: Yes

Reviewer #2: Yes

4. Is the manuscript presented in an intelligible fashion and written in standard English?

Reviewer #1: Yes

Reviewer #2: Yes

5. Review Comments to the Author

Reviewer #1: This manuscript is a potentially important addition to the existing literature on overweight and obesity in companion animals, because it confirms that some owners hold positive attitudes towards overweight in pet cats, and this is associated with increased likelihood of overweight in their own cats. If these attitudes can be changed over the long-term, it may improve welfare outcomes for many owned cats. Before I can recommend it for publication, however, there are some substantial changes that need to be made to the ms, especially in the methods section.

Major concerns:

1. The measure of cat body condition is owner-reported, which is an inherent limitation in the ms that needs to be addressed much farther up than where it's mentioned in the limitations section of the discussion. The authors attempt to measure BCS in several different ways to get around this problem, which is to their credit, but ultimately decide that the owner's own perception of BCS is the most appropriate one. Since pet owners are notoriously bad at estimating their pet's BCS, this needs to be reported all through the ms. Instead of saying that the study is looking at associations between attitudes and overweight/obesity, which implies that BCS was somehow objectively measured, it needs to be clear that the study is talking about owner-reported BCS, which is a very different thing than an objective measure. I appreciate that evaluating BCS in over 1,000 cats is not at all feasible, but this limitation needs to be clearer throughout.

2. The methods section is currently structured in a very unclear way. While I do not subscribe to the belief that all methods sections must follow the standard format of participants - materials - procedure - analysis, I do think that this particular ms would benefit greatly from either following that structure, or at a minimum, restructuring it to be more intuitive to the reader. For example, the section about association between participants' attitudes and cat BCS', much of that info really belongs nearer to the 'study design' section where the authors first mention the survey instrument. Furthermore, while I understand that the authors are attempting to be fully transparent in the way they measured BCS, the section on 'outcome variable' is way too long and confusing. It could be tightened up by at least half without losing any impact. Also, saying 'results displayed in Results section below' (L 155) strikes me as an intellectually lazy way of writing an ms - better to structure the entire piece in a way that the relevant information is presented clearly and logically, rather than expecting the reader to do the work of finding the information elsewhere.

3. How was the survey advertised? The ms indicates that it was advertised via social media, vet offices, etc, but what was the stated aim of the survey in the ad? Was it mentioned that it's about overweight and obesity in cats? Or something else? This is important information to understand whether there may have been a selection bias in respondents.

Minor comments:

The first two keywords aren't necessary because they're already in the title. Any words in the title are automatically indexed by the search engines, as I understand it.

Abstract

Add one introductory sentence at the top of the abstract explaining why we should even care about this at all.

L24-25 the range of % is unclear - where does the range come from?

L28 - mention that this is owner-reported BCS, rather than somehow objectively measured BCS.

Main text

L49 - I'm not convinced that the S1 table is strictly necessary for this report, but whether it's kept or not, a short summary should be provided in the intro. Adding a short para will be helpful for the reader to understand what's been done before. Don't expect them to find the supplementary material to get the basic info. Similarly, it's referred to again in L70, which would suggest that it's perhaps important enough to include in the main document, or just dispense with the table altogether and give a brief overview.

L91 - describe the BCS measures in more detail here. Again - it's not reasonable to expect most readers to access the supplementary files, so sufficient info should be provided in text to give the most important info.

L117 - the three instances of 'attitude' should perhaps be 'attitudes'.

L139 - should 'somehow' be 'somewhat'?

L141 - clarify 'these three candidate outcomes'

L160 - Table 1 should occur shortly after where it is first referred to in text, not several pages later.

L180 - Sentence starting with 'biologically meaningful' is unclear.

L182 - suggest changing 'half of or more than half of' to 'at least half'

L188 - why were the hunting items, specifically, selected to be the reliability items?

L265 - Suggest adding M/SD for each item in Table 2, in addition to the frequency results. Also, suggest running a PCA to reduce the number of variables needed for further analyses. These items may all load on a handful of components, which could be used to create composite variables, and potentially make your further analyses clearer conceptually.

L270 - think 'obesity' should be 'obese'

L283 - Table 3's categories 'agreeing, neutral, critical' could be better written, possibly as 'agreeing, neutral, disagreeing' or 'positive, neutral, negative'.

L331 - Table 5 - given the large amount of info here, I suggest highlighting/bold the sig results. Also suggest adding n's for each category.

L344 - same with Table 6

L389 - but BCS vet was included in an analysis, so to say it was excluded doesn't make sense.

392-392 - I don't think the authors can make this claim without some sort of objective measure of BCS. And relying on owner-reported veterinary perceptions isn't objective.

L492 - being fed 4 times/day was associated with underweight, which seems to contradict the earlier statement (L480) that cats being fed that often was associated with overweight. Please clarify.

L507-515 are unclear and would benefit from being rephrased. Also, please cite the claim on L511.

L559 - 'slight' over-representation of women? It was 85% women. Please clarify.

Reviewer #2: The manuscript “Positive attitudes towards feline obesity are strongly associated with ownership of obese cats” by Teng et al. is an investigation on risk factors for overweight and obesity in Australian cats.

The aims of the study were to investigate:

a) “Cat owner´s attitudes towards feline overweight and obesity (O&O) and their associations with O&O in their cats

b) The risk factors for feline O&O and underweight, particularly those involving owner practice

The investigation was based on an online survey targeting the Australian cat owning population and 1,390 responses were evaluated to be valid for inclusion in the statistical analyses.

Feline obesity is a major worldwide problem and unfortunately it seems to be increasing. Therefore, it is very relevant to gain information on risk factors – to better enable successful preventative strategies. There are currently several studies investigating managerial risk factors, many of these identifying neutering and indoor confinement as major contributors to feline obesity while also differences in feeding management has been identified in several studies but results are not always agreeing. Fewer studies have focused on the owners attitudes to cats, and how owners perceive feline obesity. This is relevant information as most cats rely on their owners for food and thus owners should in most cases be able to prevent development of obesity. This study identifies attitudes that the veterinary community has to address in order to combat overweight and obesity in cats.

The manuscript is interesting and generally well written.

There are however a few major issues concerning the study design that should be addressed because they could significantly influence the results. These issues cannot be changed, but the authors should emphasise these limitations and how they could affect the results.

First of all, the investigation is an online survey and therefore the parameter “body composition (BCS)” is assessed by the owner. It has been shown in several studies that owners to a very large degree overestimate the BCS of underweight cats and underestimate the BCS of overweight and obese cats. The authors are well aware of this issue and therefore try to identify alternative measures that could support owner assessed BCS. This includes veterinarian assessed BCS and bodyweight assessed BCS. Based on these data, the authors document that 21% of the cats evaluated to have a BCS4 by a veterinarian are evaluated to be normal weight by the owner. Because only 59% of the cats were evaluated by a veterinarian, the authors choose to only base the investigation on owner assessed body composition. This results in a relatively low prevalence of O&O compared with other studies. However, there is no mention in the manuscript as to how this could affect the results. It is imperative that this is discussed in more detail.

When reviewing the results on the different BCS assessments, Table 4 gives a good overview – but there is something wrong with figure 1? The numbers are reflecting different populations, the BCSvet reflect the population included in table 4 while BCS figure and BCS owner seems to reflect numbers from the full data set – this makes no sense. Either use the same dataset or exclude BCS VET, or omit the figure as the same is illustrated in table 4.

The authors write that one of the major advantages with this study is the inclusion of underweight cats. Again it should be commented that 23% of the cats that the veterinarian found to be underweight were evaluated as normal weight by the owner. The information about underweight in cats rely on 100 cats most of them senior cats with potential concomitant diseases. This population of cats seems significantly different from the O&O population and seems to reflect different issues. Therefore it seems inappropriate to handle the two populations statistically together by providing an overall P-Value in the tables (table 5 & 6). By doing this a factor such as dry food being a major part of the diet seems to become very significant while the table shows that there is no difference between dry food being a major part of the diet and not being a part of the diet.

Other comments relating to the tables and presentation of results. In the results section, provide the BCS results prior to the questionnaire results. The reader needs to understand this to undertand the questionnaire results. For supplementary tables, currently the percentages are presented as percentages ticking of that parameter across BCS. This makes it really hard to compare BCS groups. Consider presenting it as percentage of owners ticking that parameter of within the BCS group. This will make it much easier to compare the BCS groups directly.

With regards to “stay indoors”, the authors decided to pool the categories “often” and “always” this could be potentially screwing the results as many of the cats the are often indoor are also often outside on the property and sometimes outside the property – indicating that these actually have significantly larger degree of outdoor access compared with the fully indoor confined.

Minor input:

Front page: please only include words not included in the title as keywords

Line 23: what is valid responses

Line 43-49 references are lacking

Lines 70-73: difficult to read

Line 88: did you ask if the respondent was a primary caretaker?

Line 142: individual instead of focal

Line 375-286: very detailed but not very relevant

Line 408: receive discrimination?

Line 415: paws?

Line 438: what do you mean with careless attitude?

Line 447: how do you know these are healthy cats?

Line 458: please provide a reference on decreasing energy requirement – in older cats, I have been unsuccessful in finding one

Line 459: be specific and only refer to cats as they differ from dogs

Line 498: increased energy density

Line 510: please revise sentence – does not make much sense right now

Line 558: limitations should be expanded as discussed above

6. PLOS authors have the option to publish the peer review history of their article (what does this mean?). If published, this will include your full peer review and any attached files.

Reviewer #1: No

Reviewer #2: No

---

## [Author Response · Author response to Decision Letter 0]

9 Mar 2020

Dear Editor

We appreciate the feedback from the reviewers and the editor. We would like to take this opportunity to respond to your comments. 

Sincerely

Kendy

Reviewer #1: This manuscript is a potentially important addition to the existing literature on overweight and obesity in companion animals, because it confirms that some owners hold positive attitudes towards overweight in pet cats, and this is associated with increased likelihood of overweight in their own cats. If these attitudes can be changed over the long-term, it may improve welfare outcomes for many owned cats. 

Authors’ response: Thank you very much.

Before I can recommend it for publication, however, there are some substantial changes that need to be made to the ms, especially in the methods section.

Major concerns:

1. The measure of cat body condition is owner-reported, which is an inherent limitation in the ms that needs to be addressed much farther up than where it's mentioned in the limitations section of the discussion. The authors attempt to measure BCS in several different ways to get around this problem, which is to their credit, but ultimately decide that the owner's own perception of BCS is the most appropriate one. Since pet owners are notoriously bad at estimating their pet's BCS, this needs to be reported all through the ms. Instead of saying that the study is looking at associations between attitudes and overweight/obesity, which implies that BCS was somehow objectively measured, it needs to be clear that the study is talking about owner-reported BCS, which is a very different thing than an objective measure. I appreciate that evaluating BCS in over 1,000 cats is not at all feasible, but this limitation needs to be clearer throughout.

Authors’ response: We agree with the reviewer’s comments. From the outset, we had realised that we will have to deal with this problem. That is why we took a number of initiatives such as using three different measures of BCS and calculating the agreements between different BCS measures. However, as more than one-third of cats didn’t have BCS evaluated by a veterinarian, it was impossible to use it as the outcome in the models. In the previous version of the manuscript, we had acknowledged that the BCSs were evaluated by the owners, and we have now used the term ‘owner-reported BCS’, where appropriate, throughout the manuscript. 

We also added a paragraph to acknowledge the misclassification bias caused by using the owner-reported BCS, shown as below (lines:573-590): 

Secondly, the owner-reported BCS may be subject to misclassification bias. Although we made several attempts to reduce misclassification bias, some were unavoidable in the current study. As discussed above, owners often underestimate the BCS of their cats [8-10]. This manifested in the current results in that the owner-perceived BCS of their cat (i.e., BCS Owner) was idealised (i.e., reported towards BCS 3). This is to be expected because underweight or overweight cats may be considered socially undesirable. The misclassification bias could be non-differential, i.e. it may not depend on other variables. If this is the case, the bias would be towards the null, i.e. the magnitude of parameter estimates would be lower than their true values. Non-differential misclassification would have been a greater concern for the current study had we not identified any effects. However, given that we did find many significant associations, we believe that any effect of non-differential misclassification would not be severe. The differential misclassification bias is also possible in this study, i.e. the classification error is not independent but rather depends on other variables. For example, it is likely that owners who tended to report ideal BCS might also have under-reported inattentive owner practices such as providing more food than the cat needs. Differential misclassification can either exaggerate or underestimate an effect, so the findings of the current study should be interpreted with caution.

2. The methods section is currently structured in a very unclear way. While I do not subscribe to the belief that all methods sections must follow the standard format of participants - materials - procedure - analysis, I do think that this particular ms would benefit greatly from either following that structure, or at a minimum, restructuring it to be more intuitive to the reader. For example, the section about association between participants' attitudes and cat BCS', much of that info really belongs nearer to the 'study design' section where the authors first mention the survey instrument. 

Authors’ response: We have rearranged the Methods section and moved the first part of the association between the participants’ attitude towards feline O&O and their cat’s body condition to the questionnaire design and implementation section under Study Design section (lines 106 - 111). 

Furthermore, while I understand that the authors are attempting to be fully transparent in the way they measured BCS, the section on 'outcome variable' is way too long and confusing. It could be tightened up by at least half without losing any impact. 

Authors’ response: We have shortened the paragraph and moved parts of this section to Study Design. Here is the new paragraph (lines 137 – 149):

The three candidates for the cat BCS outcome variable were BCS Owner, BCS Figure, and BCS Vet. Although BCS Vet was most likely to be close to the true BCS of cats, it could not be considered because many participants (n=571, 41.1%) did not provide this information. To determine whether BCS Owner or BCS Figure was more suitable, the levels of agreement between BCS Vet and both BCS Owner and BCS Figure were evaluated separately by calculating weighted kappa using the ‘psych’ package [33]. Furthermore, the weighted kappa between categorised body weight and BCS Owner and BCS Figure were calculated for several breeds of cats whose ideal weight ranges were documented on a website named ‘Cat Owner Club’ [34]. BCS Owner [with three categories: underweight (BCS of 1 or 2), ideal weight (BCS of 3) and O&O (BCS of 4 or 5)] was chosen as the outcome variable for the analyses reported here because the values of weighted kappa between BCS Owner and both BCS Vet and categorised body weight from Cat Owner Club were higher than those between BCS Figure and these two variables.

Also, saying 'results displayed in Results section below' (L 155) strikes me as an intellectually lazy way of writing an ms - better to structure the entire piece in a way that the relevant information is presented clearly and logically, rather than expecting the reader to do the work of finding the information elsewhere.

Authors’ response: We have rewritten this paragraph, shown as below (lines 145-149):

BCS Owner [with three categories: underweight (BCS of 1 or 2), ideal weight (BCS of 3) and O&O (BCS of 4 or 5)] was chosen as the outcome variable for the analyses reported here because the values of weighted kappa between BCS Owner and both BCS Vet and categorised body weight from Cat Owner Club were higher than those between BCS Figure and these two variables.

3. How was the survey advertised? The ms indicates that it was advertised via social media, vet offices, etc, but what was the stated aim of the survey in the ad? Was it mentioned that it's about overweight and obesity in cats? Or something else? This is important information to understand whether there may have been a selection bias in respondents.

Authors’ response: Our advertisements were rather general. No overweight/obesity or underweight was mentioned. The aim mentioned: The study is to investigate the factors that may influence the body composition of cats and people’s attitudes to the compositions. We have now included a copy of the advertisement as a supplementary file (S2 File). We have also added the following sentence to the methods section (lines 117-119):

The advertisements did not mention overweight or obesity but did mention the aim of the study (S2 File).

Minor comments:

The first two keywords aren't necessary because they're already in the title. Any words in the title are automatically indexed by the search engines, as I understand it.

Authors’ response: Thank you. They have been removed.

Abstract

Add one introductory sentence at the top of the abstract explaining why we should even care about this at all.

Authors’ response: A sentence has been added, shown as below (lines 18-19):

Overweight and obesity (O&O) is a risk factor for several health conditions and can result in a shorter lifespan for cats. The objectives of this study were to investigate…

L24-25 the range of % is unclear - where does the range come from?

Authors’ response: Clarification has been added as follows (lines 25-27):

In response to ten attitude-related questions, more participants (percentage range among the ten questions: 39.1–76.6%) held a disapproving attitude towards feline O&O than a neutral (17.1–31.9%) or approving attitude (3.9–27.7%).

L28 - mention that this is owner-reported BCS, rather than somehow objectively measured BCS.

Authors’ response: Clarification has been added as follows (lines 21-24): 

An online survey comprising questions related to cat owners’ attitudes towards feline O&O, owner-reported body weight and body condition of their cat, and potential risk factors for feline O&O was conducted.

Main text

L49 - I'm not convinced that the S1 table is strictly necessary for this report, but whether it's kept or not, a short summary should be provided in the intro. Adding a short para will be helpful for the reader to understand what's been done before. Don't expect them to find the supplementary material to get the basic info. Similarly, it's referred to again in L70, which would suggest that it's perhaps important enough to include in the main document, or just dispense with the table altogether and give a brief overview.

Authors’ response: We would much prefer to retain the S1 Table and have added a paragraph to summarise this piece of information (lines 50 – 55), shown as follows:

In contrast to intrinsic risk factors that have been well documented in the past 30 years, extrinsic risk factors have shown relatively inconsistent associations. Details of risk factors investigated are shown in S1 Table. Briefly, many studies have shown that male sex [6-10], neutered cats [11-17], middle age [6, 11-13, 18-21] and mixed breed [12, 14, 19-21] are associated with an increased risk of O&O. The extrinsic risk factors supported by the best evidence include feeding dry food [9, 22] and feeding treats/table scraps [6, 9, 15].

L91 - describe the BCS measures in more detail here. Again - it's not reasonable to expect most readers to access the supplementary files, so sufficient info should be provided in text to give the most important info.

Authors’ response: The BCS measures have been added in this section (lines 97 – 104), shown as follows:

Section 2 featured questions related to the cat’s body weight and body condition. Three BCS measures were asked in this section: 1) participant-perceived BCS designated by participants based on descriptions in the questionnaire (‘BCS Owner’), 2) BCS that was determined by participants choosing from five different depictions of cat shape the one most similar to their cat’s shape (‘BCS Figure’), and 3) BCS determined by the cat’s veterinarians in the past year (‘BCS Vet’). The BCS of 1-to-5 for BCS Owner and BCS Vet were labelled as: ‘very underweight (BCS of 1)’, ‘somewhat underweight (BCS of 2)’, ‘ideal (BCS of 3)’, ‘chubby/overweight (BCS of 4)’, ‘fat/obese (BCS of 5)’. Section 3 contained questions associated with ownership…

L117 - the three instances of 'attitude' should perhaps be 'attitudes'.

Authors’ response: It has been corrected accordingly. 

L139 - should 'somehow' be 'somewhat'?

Authors’ response: It has been corrected. Thank you. Also, this section has been moved to study design section.

L141 - clarify 'these three candidate outcomes'

Authors’ response: Clarification has been provided as described below:

The three candidates for the cat BCS outcome variable were BCS Owner, BCS Figure, and BCS Vet. Although BCS Vet was most likely to be close to the true BCS of cats…

L160 - Table 1 should occur shortly after where it is first referred to in text, not several pages later.

Authors’ response: Table 1 has been moved accordingly. 

L180 - Sentence starting with 'biologically meaningful' is unclear.

Authors’ response: The sentence has been changed to:

Pairwise interactions that might be biologically meaningful were evaluated in the model and, if significant, retained.

L182 - suggest changing 'half of or more than half of' to 'at least half'

Authors’ response: It has been changed accordingly. 

L188 - why were the hunting items, specifically, selected to be the reliability items?

Authors’ response: These questions were selected because they essentially measured the same issue in different ways, which allowed us to investigate coherence in participants’ responses as an indicator of the reliability of their responses. 

L265 - Suggest adding M/SD for each item in Table 2, in addition to the frequency results. Also, suggest running a PCA to reduce the number of variables needed for further analyses. These items may all load on a handful of components, which could be used to create composite variables, and potentially make your further analyses clearer conceptually.

Authors’ response: It is unclear if the reviewer means “mean/standard deviation”? As the data are discrete, we are not able to report means and standard deviations. Thank you for the suggestion of running a PCA. We decided not to run a PCA as (a) the paper is already quite lengthy (b) we would like to have coefficients for individual questions, and (c) the data are discrete. Therefore, we would like to keep this section as is if it is okay with you. 

L270 - think 'obesity' should be 'obese'

Authors’ response: Thank you; it has been corrected. 

L283 - Table 3's categories 'agreeing, neutral, critical' could be better written, possibly as 'agreeing, neutral, disagreeing' or 'positive, neutral, negative'.

Authors’ response: We have changed all categories to approving, neutral and disapproving. 

L331 - Table 5 - given the large amount of info here, I suggest highlighting/bold the sig results. Also suggest adding n's for each category.

Authors’ response: P-values < 0.05 have been highlighted. However, n’s were not added into the table to avoid cluttering as (a) the table already has a great amount of information and (b) this information can be found in the supplementary materials. 

L344 - same with Table 6

Authors’ response: P-values < 0.05 have been highlighted.

L389 - but BCS vet was included in an analysis, so to say it was excluded doesn't make sense.

Authors’ response: The sentences have been rephrased, shown as follows:

There were three candidates for the outcome variable for BCS evaluation, namely, BCS Owner, BCS Figure and BCS Vet. We could not use BCS Vet due to a large number of missing values.

392-392 - I don't think the authors can make this claim without some sort of objective measure of BCS. And relying on owner-reported veterinary perceptions isn't objective.

Authors’ response: The sentences have been changed, shown as follows (lines 403-406):

However, in contrast to what we had presumed based on the literature, BCS Owner better reflected BCS Vet than BCS Figure. That said, we acknowledge that Vet BCS might also be biased as it was reported by the owner and not obtained directly from the veterinarian (i.e., may have been subject to recall bias). It was noted that,…

L492 - being fed 4 times/day was associated with underweight, which seems to contradict the earlier statement (L480) that cats being fed that often was associated with overweight. Please clarify.

Authors’ response: The sentences have been rephrased, as shown as follows.

The frequent feeding of these cats is likely to be an owner response to the underweight status instead of the cause of underweight.

L507-515 are unclear and would benefit from being rephrased. Also, please cite the claim on L511.

Authors’ response: The sentences have been rephrased, as shown as follows (lines 517-521). Also, L511 has been cited.

Although feeding treats and/or table scraps have been reported to relate to feline O&O [6, 9, 15], this association did not emerge in the current study. It is worth noting here that, in contrast to the current study, two of the studies that reported this association did not account for other potential risk factors associated with O&O in the analysis [6, 9, 15]. 

L559 - 'slight' over-representation of women? It was 85% women. Please clarify.

Authors’ response: As we discussed that one survey showed that 76% of cat owners in Australia were female (Animal Medicines Australia Pty Ltd. Pet ownership in Australia. 2016, line 383). However, we’ve deleted “slight” in the sentence. 

The current study has some limitations in addition to the overrepresentation of female owners and the recall bias of the information about BCS Vet.

Reviewer #2: The manuscript “Positive attitudes towards feline obesity are strongly associated with ownership of obese cats” by Teng et al. is an investigation on risk factors for overweight and obesity in Australian cats.

The aims of the study were to investigate:

a) “Cat owner´s attitudes towards feline overweight and obesity (O&O) and their associations with O&O in their cats

b) The risk factors for feline O&O and underweight, particularly those involving owner practice

The investigation was based on an online survey targeting the Australian cat owning population and 1,390 responses were evaluated to be valid for inclusion in the statistical analyses.

Feline obesity is a major worldwide problem and unfortunately it seems to be increasing. Therefore, it is very relevant to gain information on risk factors – to better enable successful preventative strategies. There are currently several studies investigating managerial risk factors, many of these identifying neutering and indoor confinement as major contributors to feline obesity while also differences in feeding management has been identified in several studies but results are not always agreeing. Fewer studies have focused on the owners attitudes to cats, and how owners perceive feline obesity. This is relevant information as most cats rely on their owners for food and thus owners should in most cases be able to prevent development of obesity. This study identifies attitudes that the veterinary community has to address in order to combat overweight and obesity in cats.

The manuscript is interesting and generally well written.

Authors’ response: Thank you for your kind words

There are however a few major issues concerning the study design that should be addressed because they could significantly influence the results. These issues cannot be changed, but the authors should emphasise these limitations and how they could affect the results.

First of all, the investigation is an online survey and therefore the parameter “body composition (BCS)” is assessed by the owner. It has been shown in several studies that owners to a very large degree overestimate the BCS of underweight cats and underestimate the BCS of overweight and obese cats. The authors are well aware of this issue and therefore try to identify alternative measures that could support owner assessed BCS. This includes veterinarian assessed BCS and bodyweight assessed BCS. Based on these data, the authors document that 21% of the cats evaluated to have a BCS4 by a veterinarian are evaluated to be normal weight by the owner. Because only 59% of the cats were evaluated by a veterinarian, the authors choose to only base the investigation on owner assessed body composition. This results in a relatively low prevalence of O&O compared with other studies. However, there is no mention in the manuscript as to how this could affect the results. It is imperative that this is discussed in more detail.

Authors’ response: We understand the limitation of the current study and have attempted to tackle this issue critically. We have included the prevalence of O&O, estimated by using BCS Owner and OCS Vet, in the manuscript (lines 259-266), shown as followed: 

Weighted kappa suggested that BCS Owner had better agreement with both BCS Vet and BCS from Cat Owner Club than BCS Figure (Table 2). Nevertheless, 20.8% (n=44) of the cats evaluated by veterinarians to have a BCS4 were considered to have a BCS of 3 by owners, indicating that owners underestimated their cats’ body condition. On the other hand, 22.2% (n=14) of cats evaluated by veterinarians to have a BCS of 2 were considered to have a BCS of 3 by the owners, indicating that owners were more likely to report ideal body condition scores. However, the prevalence of O&O estimated using BCS Owner was only slightly lower (24.2%) than the prevalence of 25.9%, calculated using BCS Vet. Both the values of weighted kappa with BCS from Cat Owner Club were low…

We have now also included a paragraph in the discussion section about how this misclassification bias could have impacted the results (lines:573-590):

Secondly, the owner-reported BCS may be subject to misclassification bias. Although we made several attempts to reduce misclassification bias, some were unavoidable in the current study. As discussed above, owners often underestimate the BCS of their cats [8-10]. This manifested in the current results in that the owner-perceived BCS of their cat (i.e., BCS Owner) was idealised (i.e., reported towards BCS 3). This is to be expected because underweight or overweight cats may be considered socially undesirable. The misclassification bias could be non-differential, i.e. it may not depend on other variables. If this is the case, the bias would be towards the null, i.e. the magnitude of parameter estimates would be lower than their true values. Non-differential misclassification would have been a greater concern for the current study had we not identified any effects. However, given that we did find many significant associations, we believe that any effect of non-differential misclassification would not be severe. The differential misclassification bias is also possible in this study, i.e. the classification error is not independent but rather depends on other variables. For example, it is likely that owners who tended to report ideal BCS might also have under-reported inattentive owner practices such as providing more food than the cat needs. Differential misclassification can either exaggerate or underestimate an effect, so the findings of the current study should be interpreted with caution.

When reviewing the results on the different BCS assessments, Table 4 gives a good overview – but there is something wrong with figure 1? The numbers are reflecting different populations, the BCSvet reflect the population included in table 4 while BCS figure and BCS owner seems to reflect numbers from the full data set – this makes no sense. Either use the same dataset or exclude BCS VET, or omit the figure as the same is illustrated in table 4.

Authors’ response: Figure 1 included the entire sample. It doesn’t seem to accord with Table 4 because 571 participants didn’t provide information about BCS Vet. Therefore, the number of BCS Owner and BCS Figure are much lower than those in Figure 1.

The authors write that one of the major advantages with this study is the inclusion of underweight cats. Again it should be commented that 23% of the cats that the veterinarian found to be underweight were evaluated as normal weight by the owner. 

Authors’ response: The percentage has been added to Table 4, and the percentage has been included in the manuscript (lines 259-265), shown as follows:

Weighted kappa suggested that BCS Owner had better agreement with both BCS Vet and BCS from Cat Owner Club than BCS Figure (Table 2). Nevertheless, 20.8% (n=44) of the cats evaluated by veterinarians to have a BCS4 were considered to have a BCS of 3 by owners, indicating that owners underestimated their cats’ body condition. On the other hand, 22.2% (n=14) of cats evaluated by veterinarians to have a BCS of 2 were considered to have a BCS of 3 by the owners, indicating that owners were more likely to report ideal body condition scores. 

The information about underweight in cats rely on 100 cats most of them senior cats with potential concomitant diseases. This population of cats seems significantly different from the O&O population and seems to reflect different issues. Therefore it seems inappropriate to handle the two populations statistically together by providing an overall P-Value in the tables (table 5 & 6). By doing this a factor such as dry food being a major part of the diet seems to become very significant while the table shows that there is no difference between dry food being a major part of the diet and not being a part of the diet.

Authors’ response: We have checked whether there were sufficient samples for each of the categories of the variables that we included in the modelling. As mentioned in lines 170-176, if more than 20% of the cells had less than five cats and if any of the cells of contingency tables contained a zero when the contingency table against outcome variable, explanatory variables were re-categorised. Also, although there is an overall P-value, there are separated P-values that will help us to see the difference in the values between underweight, normal weight and overweight cats. It is usually better to have fewer models than more if it is possible. Our approach also allows comparison between underweight and overweight cats, if that’s of the readers’ interest. We would, therefore, like to keep the analyses as they are.

Other comments relating to the tables and presentation of results. In the results section, provide the BCS results prior to the questionnaire results. The reader needs to understand this to undertand the questionnaire results. 

Authors’ response: The BCS results have been moved as suggested.

For supplementary tables, currently the percentages are presented as percentages ticking of that parameter across BCS. This makes it really hard to compare BCS groups. Consider presenting it as percentage of owners ticking that parameter of within the BCS group. This will make it much easier to compare the BCS groups directly.

Authors’ response: We understand your concerns. However, we have presented the row percentages, i.e. how the distribution of cats with different BCS vary among different subgroups. Column percentages are more commonly used for case-control studies, i.e. if we had specifically selected low and high BCS cats. The row percentages are more commonly used for the cross-sectional study presented here and therefore, we would like to keep this table as is. 

With regards to “stay indoors”, the authors decided to pool the categories “often” and “always” this could be potentially screwing the results as many of the cats the are often indoor are also often outside on the property and sometimes outside the property – indicating that these actually have significantly larger degree of outdoor access compared with the fully indoor confined.

Authors’ response: We understand your concern. However, we categorised variables of the same type in the same way to avoid arbitrariness as otherwise, we could be criticised for not adhering to a set categorisation plan. Therefore, we would like to keep the original categorisation. 

Minor input:

Front page: please only include words not included in the title as keywords

Authors’ response: This has been corrected. 

Line 23: what is valid responses

Authors’ response: They are the responses with answers to at least one of the questions about the evaluation of the BCS of cats

Line 43-49 references are lacking

Authors’ response: This paragraph has been extended, and references have been added (lines 50-55). Please see as follows:

In contrast to intrinsic risk factors that have been well documented in the past 30 years, extrinsic risk factors have shown relatively inconsistent associations. Details of risk factors investigated are shown in S1 Table. Briefly, many studies have shown that male sex [6-10], neutered cats [11-17], middle age [6, 11-13, 18-21] and mixed breed [12, 14, 19-21] are associated with an increased risk of O&O. The extrinsic risk factors supported by the best evidence include feeding dry food [9, 22] and feeding treats/table scraps [6, 9, 15].

Lines 70-73: difficult to read

Authors’ response: This sentence has been rephrased as follows (lines 75-77): 

To provide greater evidence for potential extrinsic risk factors for cat O&O, and to improve our understanding of the attitudes towards feline O&O among cat owners and its associations with their cats’ body condition, the current study was conducted. Specifically, it investigated (a) the risk factors for feline O&O and underweight,…

Line 88: did you ask if the respondent was a primary caretaker?

Authors’ response: We did not ask this question in the questionnaire. This has been acknowledged in the manuscript now (lines 571-573) and shown as follows:

Firstly, we did not ask whether the participants were the primary caretaker of the cat. This might increase the chance of acquiring inaccurate information if some participants were not the primary caretakers. Secondly,…

Line 142: individual instead of focal

Authors’ response: This has been changed accordingly. 

Line 375-286: very detailed but not very relevant

Authors’ response: This paragraph has been shortened and shown as follows (lines 391-397):

In the current study, most cats were neutered (97.8%), and the percentage of neutering was higher than the overall Australian population (89%) [42]. Twenty per cent of cats in the current study population was purebred or pedigree, slightly lower than two other Australian-wide statistics at 23.4% [46] and 24% [42] and New South Wales Companion Animals Register (22.4%) [48]. Although Australian council registration rates and cat health insurance rates were estimated at 72% and 19% in 2016 [42], respectively, they were only 63.1% and 11.3% in the current study.

Line 408: receive discrimination?

Authors’ response: This has been deleted.

Line 415: paws?

Authors’ response: This has been changed accordingly. 

Line 438: what do you mean with careless attitude?

Authors’ response: This has been changed to “an indifferent attitude” as shown as below (lines 448-451):

This might be explained by the finding that a greater proportion of respondents agreed that ‘being fat/obese is undesirable’ than ‘being chubby/overweight is undesirable’. So, the questions about attitudes towards chubbiness are likely to be more sensitive to reflecting an indifferent attitude towards O&O among cat owners.

Line 447: how do you know these are healthy cats?

Authors’ response: We do not think these are healthy cats. Here we only describe the approach of some studies. 

Line 458: please provide a reference on decreasing energy requirement – in older cats, I have been unsuccessful in finding one

 Authors’ response: The reference has been provided in the manuscript. 

Laflamme D, Gunn-Moore D. Nutrition of aging cats. Veterinary Clinics of North America: Small Animal Practice. 2014;44(4):761-74. doi: https://doi.org/10.1016/j.cvsm.2014.03.001.

Line 459: be specific and only refer to cats as they differ from dogs

Authors’ response: This sentence has been changed as follows:

However, a cat’s ability to digest fat (and potentially protein) also decreases with age, resulting in reduced energy absorption [67].

Line 498: increased energy density

Authors’ response: This has been changed accordingly. 

Line 510: please revise sentence – does not make much sense right now

Authors’ response: This has been changed, shown as follows (lines 522-523):

The amount of food fed to the cats as guided by their apparent appetite was shown to be associated with a lower BCS in the current study.

Line 558: limitations should be expanded as discussed above

Authors’ response: The section of limitation has been expanded, shown as previously stated.

---

## [Decision Letter · Decision Letter 1]

7 Apr 2020

PONE-D-19-30666R1

Positive attitudes towards feline obesity are strongly associated with ownership of obese cats

PLOS ONE

Dear Dr Teng,

Thank you for submitting your manuscript to PLOS ONE. After careful consideration, we feel that it has merit but does not fully meet PLOS ONE’s publication criteria as it currently stands. Therefore, we invite you to submit a revised version of the manuscript that addresses the points raised during the review process.

In addition to the reviewer comments listed below, I have the following comments after my own thorough reading of the paper:

Line 38: Please avoid expressions such as "positively associated with", especially in the abstract, as it is ambiguous and may be interpreted as if saying that the association is of a desirable nature. 

Line 97: Please explain the acronym BCS the first time you use it.

Line 118: If the adverts did not mention obesity or overweight, how did they mention the aim of the study?

Line 243: "weighted" should be "weighed" (assuming that it refers to measuring the weight)

Line 327: What does "included an increase in age before 11 years" mean? Consider rephrasing.

Lines 589-590: Can you please provide an example of, or consider rephrasing, "logistic regression cannot account for bidirectional causality between explanatory and outcome variables", as this statement is difficult to understand for readers who don't have a special interest in statistical methods.

We would appreciate receiving your revised manuscript by May 22 2020 11:59PM. To enhance the reproducibility of your results, we recommend that if applicable you deposit your laboratory protocols in protocols.io, where a protocol can be assigned its own identifier (DOI) such that it can be cited independently in the future. For instructions see: http://journals.plos.org/plosone/s/submission-guidelines#loc-laboratory-protocols

We look forward to receiving your revised manuscript.

Kind regards,

I Anna S Olsson, Ph.D.

Academic Editor

PLOS ONE

Reviewers' comments:

Reviewer's Responses to Questions

**Comments to the Author**

1. If the authors have adequately addressed your comments raised in a previous round of review and you feel that this manuscript is now acceptable for publication, you may indicate that here to bypass the “Comments to the Author” section, enter your conflict of interest statement in the “Confidential to Editor” section, and submit your "Accept" recommendation.

Reviewer #1: All comments have been addressed

Reviewer #2: (No Response)

2. Is the manuscript technically sound, and do the data support the conclusions?

Reviewer #1: Yes

Reviewer #2: Yes

3. Has the statistical analysis been performed appropriately and rigorously? 

Reviewer #1: Yes

Reviewer #2: Yes

4. Have the authors made all data underlying the findings in their manuscript fully available?

Reviewer #1: Yes

Reviewer #2: Yes

5. Is the manuscript presented in an intelligible fashion and written in standard English?

Reviewer #1: Yes

Reviewer #2: Yes

6. Review Comments to the Author

Reviewer #1: Nice work. The manuscript is much improved, with all comments answered. Happy to accept it for publication now.

Reviewer #2: The manuscript “Positive attitudes towards feline obesity are strongly associated with ownership of obese cats” by Teng et al. is an investigation on risk factors for overweight and obesity in Australian cats.

The aims of the study were to investigate:

a) “Cat owner´s attitudes towards feline overweight and obesity (O&O) and their associations with O&O in their cats

b) The risk factors for feline O&O and underweight, particularly those involving owner practice

The investigation was based on an online survey targeting the Australian cat owning population and 1,390 responses were evaluated to be valid for inclusion in the statistical analyses.

Feline obesity is a major worldwide problem and unfortunately it seems to be increasing. Therefore, it is very relevant to gain information on risk factors – to better enable successful preventative strategies. There are currently several studies investigating managerial risk factors, many of these identifying neutering and indoor confinement as major contributors to feline obesity while also differences in feeding management has been identified in several studies but results are not always agreeing. Fewer studies have focused on the owners attitudes to cats, and how owners perceive feline obesity. This is relevant information as most cats rely on their owners for food and thus owners should in most cases be able to prevent development of obesity. This study identifies attitudes that the veterinary community has to address in order to combat overweight and obesity in cats.

The manuscript is interesting and generally well written and the author has addressed most of the reviewer comments satisfactorily, improving the manuscript.

There are however, a few minor comments that could improve the clarity further, these comments refer to the manuscript version with track changes:

Abstract Line 36 and manuscript line 394: would it not be more correct to classify this risk factor as middle age as you do in the discussion ?

Line 83: for owner assessed feline O&O

Line 157: their cats owner estimated body condition

Line 164-165: cats owner assessed body condition

Line 282: I am not sure the correction to weighted is right

Line 394: Fed twice daily

Line 401-404: I am not sure I understand the meaning with this sentence

Line 484: BCS Vet

Table 2: could the table legend be revised to The tabulated body condition score (BCS) evaluation by the cats veterinarian and weighted kappa between different owner reported BCS estimations based on data collected by an Australian-based online performed in 2016 (n=XXX)

Table 3, 4, 5 & 6: Please make sure to include in the table and figure legends both the wording Owner assessed body condition an well as the number of cats included,

7. PLOS authors have the option to publish the peer review history of their article (what does this mean?). If published, this will include your full peer review and any attached files.

Reviewer #1: No

Reviewer #2: No

---

## [Author Response · Author response to Decision Letter 1]

15 May 2020

Dear Editor

We appreciate the feedback from the reviewers and the editor. We would like to take this opportunity to respond to your comments. Our responses are in italics and any new text is in bold. The lines indicated here are referred to the version with tracked changes. 

We would be happy to provide any further information or clarification if required. 

Sincerely

Kendy

Dear Dr Teng,

Thank you for submitting your manuscript to PLOS ONE. After careful consideration, we feel that it has merit but does not fully meet PLOS ONE’s publication criteria as it currently stands. Therefore, we invite you to submit a revised version of the manuscript that addresses the points raised during the review process.

In addition to the reviewer comments listed below, I have the following comments after my own thorough reading of the paper:

Line 38: Please avoid expressions such as "positively associated with", especially in the abstract, as it is ambiguous and may be interpreted as if saying that the association is of a desirable nature.

Authors’ response: We have modified accordingly across the manuscript. This sentence has been changed as: Being over 11 years, receiving no dry food and receiving measured amounts of feed were associated with an increased odds of underweight in cats (Lines 36-37).

Line 97: Please explain the acronym BCS the first time you use it.

Authors’ response: This has been mentioned in Line 62. 

Line 118: If the adverts did not mention obesity or overweight, how did they mention the aim of the study?

Authors’ response: The aim mentioned on the ads is: The study is to investigate the factors that may influence the body composition of cats and people’s attitudes to the compositions.

Line 243: "weighted" should be "weighed" (assuming that it refers to measuring the weight)

Authors’ response: This has been modified accordingly (Line 244). 

Line 327: What does "included an increase in age before 11 years" mean? Consider rephrasing.

Authors’ response: This has been changed to “being middle-aged” (Line 329).

Lines 589-590: Can you please provide an example of, or consider rephrasing, "logistic regression cannot account for bidirectional causality between explanatory and outcome variables", as this statement is difficult to understand for readers who don't have a special interest in statistical methods.

Authors’ response: We have rephrased this sentence and added an example as follows: Lastly, the position of variables in logistic regression cannot imply the direction of causality. For example, in our results, frequent feeding is likely to be an owner's response to their cat’s underweight instead of the cause of underweight (Lines 592-595).

Reviewer #1: Nice work. The manuscript is much improved, with all comments answered. Happy to accept it for publication now.

Authors’ response: Thank you.

Reviewer #2: The manuscript “Positive attitudes towards feline obesity are strongly associated with ownership of obese cats” by Teng et al. is an investigation on risk factors for overweight and obesity in Australian cats.

The aims of the study were to investigate:

a) “Cat owner´s attitudes towards feline overweight and obesity (O&O) and their associations with O&O in their cats

b) The risk factors for feline O&O and underweight, particularly those involving owner practice

The investigation was based on an online survey targeting the Australian cat owning population and 1,390 responses were evaluated to be valid for inclusion in the statistical analyses.

Feline obesity is a major worldwide problem and unfortunately it seems to be increasing. Therefore, it is very relevant to gain information on risk factors – to better enable successful preventative strategies. There are currently several studies investigating managerial risk factors, many of these identifying neutering and indoor confinement as major contributors to feline obesity while also differences in feeding management has been identified in several studies but results are not always agreeing. Fewer studies have focused on the owners attitudes to cats, and how owners perceive feline obesity. This is relevant information as most cats rely on their owners for food and thus owners should in most cases be able to prevent development of obesity. This study identifies attitudes that the veterinary community has to address in order to combat overweight and obesity in cats.

The manuscript is interesting and generally well written and the author has addressed most of the reviewer comments satisfactorily, improving the manuscript.

There are however, a few minor comments that could improve the clarity further, these comments refer to the manuscript version with track changes:

Abstract Line 36 and manuscript line 394: would it not be more correct to classify this risk factor as middle age as you do in the discussion ?

Authors’ response: We have now modified accordingly (Lines 35 and 329) . 

Line 83: for owner assessed feline O&O

Authors’ response: We have modified accordingly (Line 79).

Line 157: their cats owner estimated body condition

Authors’ response: We have modified accordingly (Line 151).

Line 164-165: cats owner assessed body condition

Authors’ response: We have modified accordingly (Line 155).

Line 282: I am not sure the correction to weighted is right

Authors’ response: We have changed them back to “weighed” (Line 244).

Line 394: Fed twice daily 

Authors’ response: We have modified accordingly (Line 330).

Line 401-404: I am not sure I understand the meaning with this sentence

Authors’ response: It has been deleted as it was not supposed to be there. 

Line 484: BCS Vet

Authors’ response: We have modified accordingly (Line 406).

Table 2: could the table legend be revised to The tabulated body condition score (BCS) evaluation by the cats veterinarian and weighted kappa between different owner reported BCS estimations based on data collected by an Australian-based online performed in 2016 (n=XXX)

Authors’ response: We have now added the number of participants in the legend. We did not include “evaluation by the cats veterinarian” as it was also owner-reported. The title is now: The tabulated body condition score (BCS) evaluations and weighted kappa between different owner-reported BCS based on data collected by an Australian-based online survey in 2016 (n=1,390).

Table 3, 4, 5 & 6: Please make sure to include in the table and figure legends both the wording Owner assessed body condition an well as the number of cats included,

 Authors’ response: We have followed the instruction for the title of tables where the information of owner-assessed BCS was included.

---

## [Editor Report · Decision Letter 2]

21 May 2020

Positive attitudes towards feline obesity are strongly associated with ownership of obese cats

PONE-D-19-30666R2

Dear Dr. Teng,

We are pleased to inform you that your manuscript has been judged scientifically suitable for publication and will be formally accepted for publication once it complies with all outstanding technical requirements.

With kind regards,

I Anna S Olsson, Ph.D.

Academic Editor

PLOS ONE
---

## [Editor Report · Acceptance letter]

16 Jun 2020

PONE-D-19-30666R2 

Positive attitudes towards feline obesity are strongly associated with ownership of obese cats 

Dear Dr. Teng:

I'm pleased to inform you that your manuscript has been deemed suitable for publication in PLOS ONE. Congratulations! Your manuscript is now with our production department. 

Kind regards, 

on behalf of

Dr I Anna S Olsson 

Academic Editor

PLOS ONE